# Monovalent anion-selective membranes fabricated via in situ interfacial polymerization

Noor Ul Afsar[1], Michael Holmboe[1], C. André Ohlin [1], Niaz Ali Khan[2], Liang Ge [3] ✉, Tongwen Xu [3] & Naser Tavajohi [1] ✉

Developing monovalent anion-selective membranes (MAPMs) faces challenges, including the trade-off between flux and selectivity, membrane stability, and cost-effective fabrication. Overcoming these requires advanced material design and scalable techniques. Here, we introduce in situ interfacial polymerization (ISIP) to prepare MAPMs. Base membranes are synthesized via superacid polymerization and modified with anion channels and -NH$_2$ groups. During ISIP, trimesoyl chloride reacts with surface -NH$_2$ groups, forming a partially crosslinked structure with -COOH groups to regulate ion transport via electrostatic interactions. This results in low membrane resistance (4.7 Ω cm$^2$) and selective transport of weakly hydrated ions (Cl$^-$, Br$^-$, NO$_3^-$), while strongly hydrated ions (SO$_4^{2-}$, F$^-$) face higher barriers. MAPMs demonstrate high performance, achieving a limiting current density (>90 mA cm$^{-2}$), Cl$^-$ flux (1.98 mol m$^{-2}$ h$^{-1}$ at 5 mA cm$^{-2}$), and selectivity (244 for Cl$^-$/SO$_4^{2-}$), confirming effective hydration dynamics control and balanced performance. Simulations reveal how charge distribution affects ion migration pathways.

Global wastewater volumes are projected to rise by ~25% by 2030 and up to 50% by 2050, driven by rapid population growth, urbanization, and industrial expansion. Key sectors such as agriculture, textiles, food processing, pharmaceuticals, and automotive manufacturing contribute heavily to both water consumption and pollution[1]. A major concern in industrial wastewater is the presence of harmful ions like nitrate (NO$_3^-$), sulfate (SO$_4^{2-}$), and bromide (Br$^-$), which pose serious risks to water quality, aquatic life, soil health, and infrastructure through corrosion and ecological imbalance[2]. These growing challenges highlight the urgent need for improved separation and purification technologies.

Traditional wastewater treatment methods, such as precipitation, ion exchange, filtration, adsorption, and thermal-based approaches like membrane distillation, have been widely used[3–7]. However, these methods come with limitations, including high costs, chemical dependency, complex equipment, and the production of toxic byproducts. To improve wastewater treatment efficiency, it is essential to adopt advanced technologies that balance technical feasibility, cost-effectiveness, and environmental sustainability. One promising solution is electrodialysis (ED) with ion-selective membranes, which can efficiently separate and recover specific ions from industrial wastewater[8]. By leveraging selective ion transport properties, ED systems improve water quality and enable resource recovery, making them a sustainable alternative. However, current ion-exchange membranes in ED still face significant challenges. A major issue with traditional membranes is the trade-off between ion flux and selectivity, which results from structural limitations[9]. Conventional membranes lack precise control over ion size, charge distribution, and hydrophilicity. Consequently, selectivity remains low, and energy consumption increases[10]. Additionally, the lack of well-defined ion transport pathways in these membranes hinders efforts to analyze and optimize performance.

[1]Department of Chemistry, Umeå University, Umeå, Sweden. [2]Interdisciplinary Research Center for Membranes and Water Security, King Fahd University of Petroleum & Minerals, Dhahran, Saudi Arabia. [3]State Key Laboratory of Precision and Intelligent Chemistry, School of Chemistry and Materials Science, University of Science and Technology of China, Hefei, Anhui, China. ✉e-mail: geliang@ustc.edu.cn; naser.tavajohi@umu.se

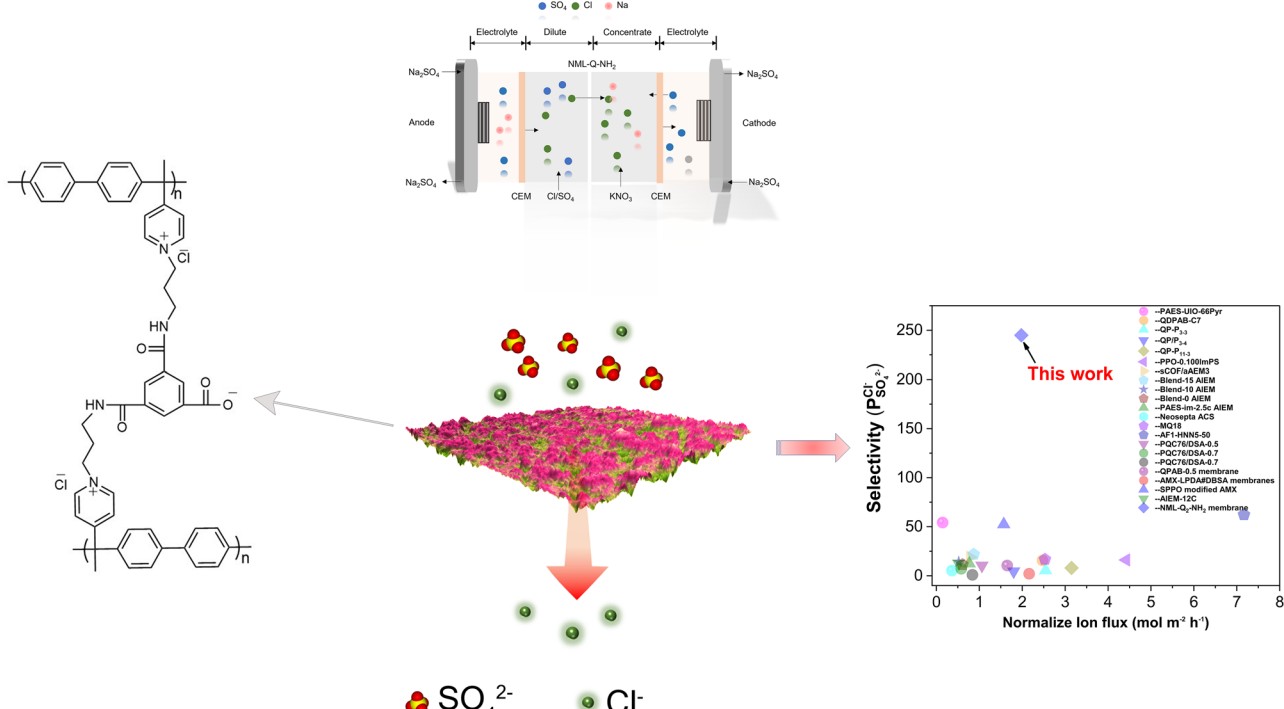

**Fig. 1 | A schematic shows a specialized polymer membrane fabricated via in-situ interfacial polymerization (ISIP) for selective monovalent ion separation using electrodialysis.** Under a current density of 5 mA cm$^{-2}$ and an equimolar 0.1 mol L$^{-1}$ NaCl/Na$_2$SO$_4$ solution, the 7.03 cm$^2$ membrane leverages electrostatic repulsion to enhance ion selectivity and separation efficiency, offering a promising approach for advanced desalination and ion separation.

To address these limitations, research has focused on developing advanced anion-selective membranes by incorporating functional materials such as zeolites[11], metal-organic frameworks (MOFs)[12], and covalent organic frameworks (COFs)[13]. While these materials show promise due to their ion selectivity and transport efficiency, challenges such as high synthesis costs, scalability, and instability in aqueous environments hinder their widespread use. For example, COFs require expensive raw materials and have scalability issues, while MOFs, despite their high selectivity, tend to be unstable in water-based applications[14]. Among polymeric membranes, anion exchange membranes (AEMs) play a critical role in selectively transporting anions, but enhancing their performance remains a complex challenge. The efficiency of AEMs largely depends on the chemical nature of the cationic head groups, their molecular structure, and how they are anchored to the polymer backbone. These factors influence the formation and stability of ion-conducting channels essential for membrane function. However, traditional AEMs are often plagued by excessive swelling when exposed to water, which can lead to pore collapse, reduced ion selectivity, and compromised mechanical stability[15]. To address these issues, researchers have explored several strategies such as cross-linking polymer chains, blending different polymers, forming acid-base pairs, and applying surface modifications[16]. Despite these efforts, none of these methods has successfully addressed the key challenges of achieving high selectivity, operational stability, and practical scalability together. While some techniques offer improvements in specific areas, a complete solution that meets both performance and manufacturing needs is still lacking. This highlights the need for a more integrated approach to membrane design that not only focuses on lab performance but also considers future application, cost, and scale-up challenges.

Here, we show that in-situ interfacial polymerization (ISIP) is a one-step, cost-efficient method for fabricating ion-selective membranes through integrated synthesis, eliminating the need for post-treatment. By incorporating design elements such as an ether-free polymer backbone, grafted quaternary ammonium groups, terminal -NH$_2$ groups for polyamide stabilization, and hydrolyzed -COOH groups for enhanced anion selectivity, ISIP overcomes key limitations of conventional approaches (Figs. 1 and 2a). The resulting membranes exhibit low resistance, rapid ion transport, and remarkable selectivity for weakly hydrated ions such as Cl$^-$, Br$^-$, and NO$_3^-$ over strongly or moderately hydrated ions like SO$_4^{2-}$ and F$^-$, driven by electrostatic interactions. These membranes achieve high limiting current density >90 mA cm$^{-2}$, significant Cl$^-$ flux, i.e., 1.98 mol m$^{-2}$ h$^{-1}$ at 5 mA cm$^{-2}$ and a high selectivity of 244 for Cl$^-$/SO$_4^{2-}$ system, confirming effective hydration control and robust performance for sustainable ion separation.

## Results

### NML-Q membranes for ion separation

The NML membranes were synthesized by functionalizing a hydrophobic PAB-based backbone with varying amounts of amines (Table S1), thereby introducing QA groups to enhance both the IEC and membrane hydrophilicity. The chemical structure of the NML-Qx polymer, in comparison with the PAB backbone, was confirmed through analysis of their $^1$H NMR and FTIR spectra (Figs. 2b, c and S1, S2). The pristine NML membrane, lacking QA groups, displayed negligible WU and was therefore characterized only after functionalization. As the amount of amine to PAB increases from 0.55 mmol to 1.10 mmol, the IEC improves from 0.35 mmol g$^{-1}$ to 0.62 mmol g$^{-1}$, demonstrating the successful grafting of QA groups. This increase in IEC is accompanied by a significant rise in WU values, from 4.21% to 16.96% and further to 30.04%, indicating enhanced hydrophilicity. After surface modifications, the IEC values showed an overall decreasing trend, except for NML-Q$_3$-NH$_2$[17]. However, in this case, the reduction is more pronounced due to the formation of covalent bonds during surface crosslinking and the generation of -COOH groups. These -COOH groups serve as active functional sites, reducing IEC through electrostatic interactions with the oppositely charged QA groups, while also causing a slight increase in WU. The process for

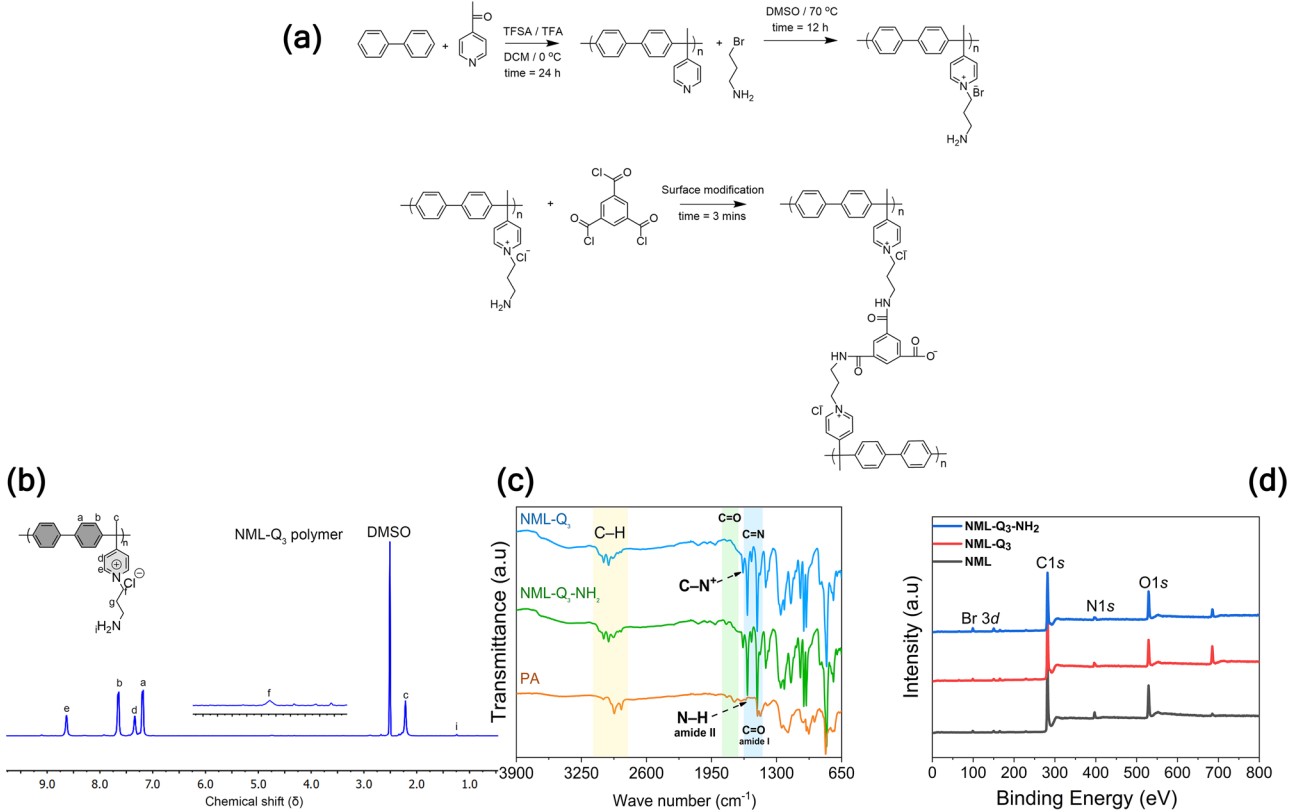

**Fig. 2 | Membrane synthesis and characterization. a** Synthetic pathway for PAB polymer, NML-$Q_x$ membranes, and NML-$Q_x$-$NH_2$ membranes. Membranes (Cl⁻ form) were subjected to ISIP surface modification, which occurs only on the membrane surface while the internal structure remains unchanged. **b** ¹H NMR spectra of the synthesized polymer (solvent: DMSO-$d_6$, 400 MHz), with detailed peak assignments corresponding to specific chemical structures, confirming the successful incorporation of desired functional groups. **c** ATR-FTIR spectra highlighting the key components of the quaternized membranes, including NML-$Q_3$, NML-$Q_3$-$NH_2$, and the polyamide (PA) layer. The spectra provide evidence of successful functionalization and chemical modification at the membrane surface. **d** XPS survey spectra displaying the surface chemical composition of the NML, NML-$Q_3$, and NML-$Q_3$-$NH_2$ membranes, confirming the presence and distribution of quaternized groups and amine functionalization on the membrane surface.

making NML-$Q_x$ membranes was limited to three membranes based on IEC and WU. Higher IEC raised WU, and too much WU could reduce selectivity. Therefore, it is crucial to maintain a careful balance between IEC and WU when designing ion-selective membranes[18]. We also tested the transport number ($t_{Cl^-}$), membrane resistance ($R_M$), and limiting current density (LCD), correlating with membrane structure to optimize ED performance (Table S2). For example, the $R_M$ of the NML-$Q_2$ and NML-$Q_3$ membranes decreased from 3.82 Ω cm² to 2.31 Ω cm², indicating improved conductivity due to increased IECs, which also resulted in an increase in the $t_{Cl^-}$ from 0.76 to 0.88. Despite the exclusion of NML-$Q_1$ and NML-$Q_1$-$NH_2$ due to low IECs, the other base and modified membranes exhibited high LCD, enhancing ion flux at high current densities[19-22]. Based on the assessment of physical characteristics provided in Fig. S3, and Table S2, the NML-$Q_2$-$NH_2$ membrane exhibited an optimal balance of high flux and selectivity. The membrane's enhanced performance originates from its three-phase structure, comprising a hydrophobic region (PAB), an ionic phase (QA groups), and an auxiliary phase. In comparison, conventional ion exchange membranes (IEMs) typically exhibit a two-phase structure, consisting of a hydrophilic domain containing functional groups and a hydrophobic polymer backbone (Fig. S4). The inclusion of the auxiliary phase fundamentally alters ion transport behavior. In standard IEMs, selectivity is achieved by allowing counterions to migrate through the hydrophilic phase, while co-ions are excluded due to electrostatic repulsion. Increasing the density of charged groups in conventional IEMs to enhance ion flux often results in excessive water uptake (WU) and swelling, which compromises membrane integrity and reduces

selectivity leading to a trade-off between flux and rejection. In contrast, the auxiliary phase introduced in this study, composed of -$NH_2$ groups, promotes microphase separation within the membrane, facilitating the transport of small ions while hindering the passage of larger ones. This structural optimization enables high selectivity without sacrificing ion flux[23,24]. The presence of alkyl spacers is essential in this structure; without them, the Menshutkin reaction would generate only QA groups (hydrophilic phase). Balancing hydrophobic phase and ionic phase is vital, as excessive WU in the active may reduce selectivity. Therefore, careful control over modifications to membrane structure is essential for achieving optimal IEC, WU, and ion selectivity[25].

## Characterization of TMC-modified membranes

To assess the surface properties of the TMC-modified membranes, XPS was employed, which provides an effective detection depth of approximately 10 nm[26]. Representative results for three membranes (NML, NML-$Q_3$, and NML-$Q_3$-$NH_2$) are presented in Fig. 2d. The presence of main elements such as C1$s$, N1$s$, Br 3$d$, and O1$s$ peaks observed at binding energies of 284 eV, 398.8 eV, and 530.7 eV, respectively[27]. Comparative analysis of these peaks in the modified membranes is summarized in Table S3 and Table S4, with detailed descriptions of element linkages within the membrane structure. The successful incorporation of amine groups through the quaternization process is evidenced by the increase in nitrogen content (0.36%, 0.48%, and 1.58% for NML-$Q_1$, NML-$Q_2$, and NML-$Q_3$, respectively) as indicated by the N1s peak at 398.8 eV, and the corresponding decrease in oxygen content

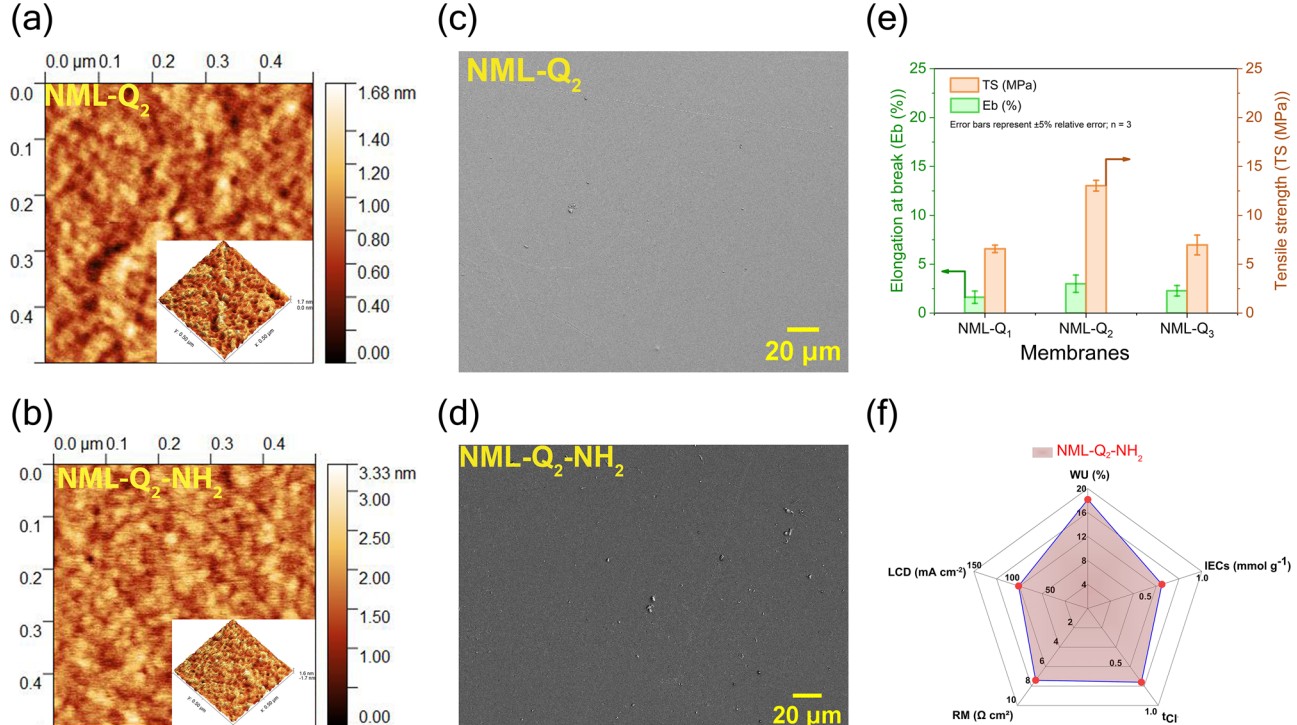

**Fig. 3 | Characterization of NML-$Q_2$ and NML-$Q_2$-$NH_2$ membranes. a, b** AFM images, along with 3D surface insets, reveal distinct surface morphologies with nanoscale hydrophilic and hydrophobic domains. The NML-$Q_2$-$NH_2$ membrane exhibits increased surface roughness (1.887 nm) compared to NML-$Q_2$ (1.037 nm), indicating enhanced surface features post-modification. **c, d** SEM images show smooth, uniform surfaces for both membranes, with no visible defects or agglomeration, confirming the precision of the fabrication process. **e** Mechanical properties of NML-$Q_x$ membranes: tensile strength (TS) and elongation at break (Eb) of NML-$Q_2$-$NH_2$ demonstrate improved mechanical integrity with higher TS (13 MPa) and moderate flexibility (Eb 3%) compared to the unmodified membranes. **f** Radar plot summarizes key performance metrics for NML-$Q_2$-$NH_2$, including WU, IEC, $t_{Cl^-}$, $R_M$, and LCD. The balanced properties enable a combination of mechanical strength, ionic conductivity, and stability, making it suitable for electrochemical applications.

(14.78%, 13.92%, and 10.92%), which aligns with the enhanced IEC values of the membranes.

Additionally, AFM and SEM analysis were conducted to investigate the phase separation, surface roughness, and morphology of the prepared membranes, as shown in Figs. 3a–d and S5–S7. The membranes also appeared visually transparent and flexible, highlighting their practical handling and applicability (Fig. S8). The AFM images revealed distinct hydrophilic and hydrophobic domains, with dark regions representing hydrophilic domains and bright regions corresponding to hydrophobic domains. These hydrophilic regions, penetrating through the membrane, form ion channels essential for ion transport, while the hydrophobic domains contribute to structural stability and mechanical strength[28,29]. The phase separation observed in the NML-$Q_x$ membranes results from the interaction between the rigid aromatic backbone, pyridinium cation side chains, and flexible aliphatic filaments linked by -$NH_2$ groups, which contribute to the distinct morphology of the membrane[28]. The surface roughness of the membranes was also measured to evaluate the changes induced by the ISIP process. The increase in roughness observed aligns with the potential benefits of enhanced permeability, as higher surface roughness increases the active surface area[30,31]. However, increased roughness also presents a potential drawback: membranes with greater roughness are more susceptible to fouling[32]. The surface roughness ranged from 1.032 nm to 1.139 nm, increasing slightly to 1.33–2.024 nm after surface modification, which is consistent with expectations for IP processes[30,33]. Notably, the roughness values observed in this study are lower than those typically reported for traditional IP processes, where the addition of a secondary monomer often results in greater surface irregularities. This relatively lower roughness is a positive aspect of the ISIP process, balancing improved permeability with reduced fouling risk.

The SEM images showing that the membranes have a smooth and uniform morphology with minimal surface agglomeration.

The structural integrity is further supported by the mechanical properties, including tensile strength (TS) and elongation at break (Eb), which showed a non-uniform trend with WU (Fig. 3e). While both TS and Eb varied across the NML-Q membranes, alternating between increases and decreases with changes in hydration levels, the values remained sufficiently high to ensure mechanical stability for demanding applications like ED. For instance, the representative membrane, i.e., NML-Q2, exhibited the highest TS (13 MPa) with Eb (3%). These results suggest that, despite minor fluctuations, the membranes maintain toughness, and dimensional stability, along with a balanced set of performance metrics summarized in the radar plot (Fig. 3f), including WU, IEC, $t_{Cl^-}$, RM, and LCD, all of which are required for high-duty ED processes.

Expanding on the structural and mechanical evaluations, we further studied the solubility of the membranes in various solvents, including DMF, DMSO, THF, DMAc, and NMP. Interestingly, the membrane (NML-$Q_2$-$NH_2$, 60 mg/3 mL) showed complete solubility at room temperature in all solvents except THF (Fig. S9 and Table S5). This solubility profile reflects the processability and scalability of the membrane fabrication (Fig. S10).

## Electrokinetic and conductivity analysis

The zeta (ζ) potential, a crucial factor in determining membrane permselectivity, was measured for all membranes both before and after modification with TMC at pH 7.1, the relevant pH for the ED process of this study. The ζ potential increases from +1.3 mV to +3.74 mV, indicating enhanced positive charge due to the presence of QA groups (Fig. S11). However, modification with TMC and subsequent

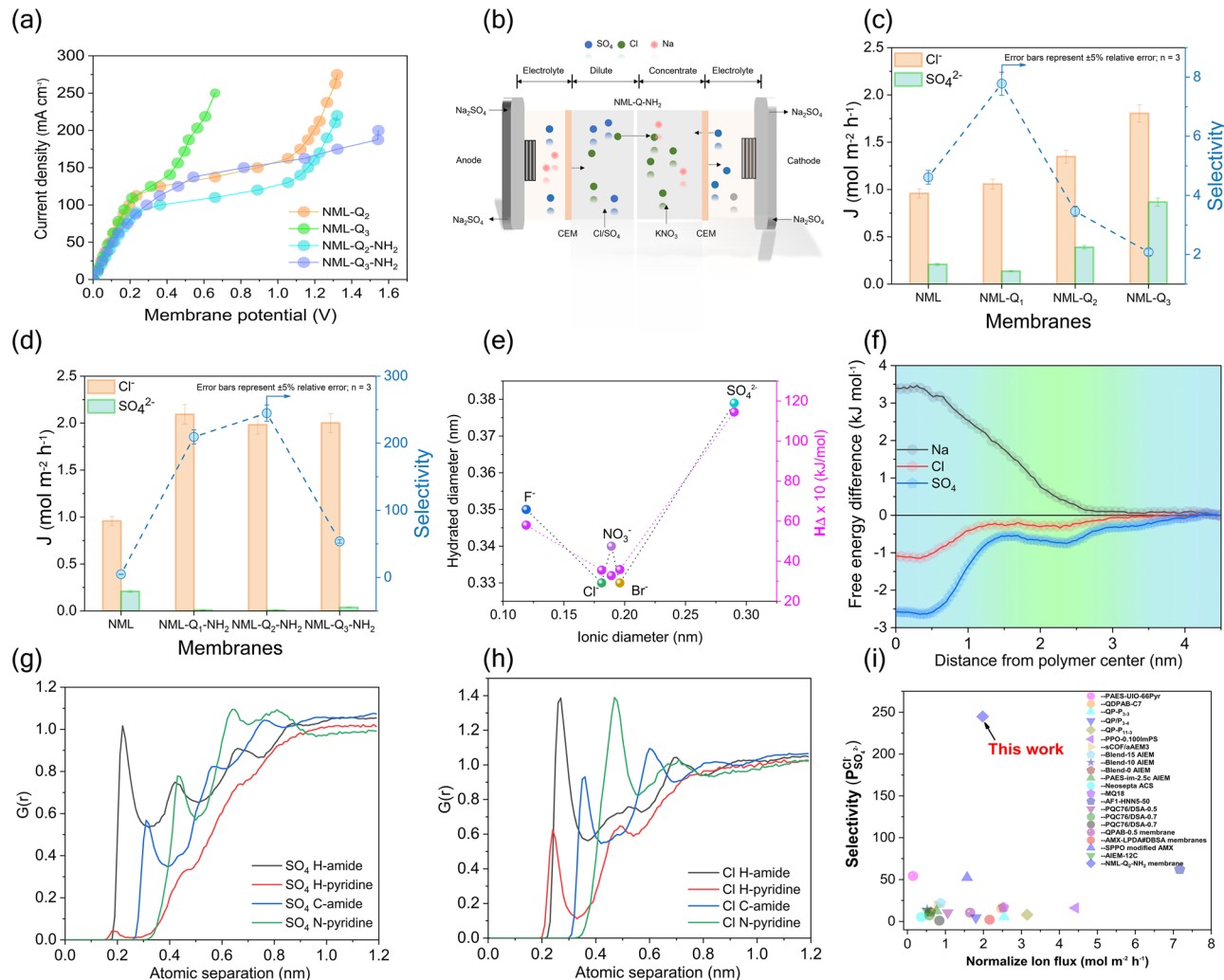

**Fig. 4 | Ion transport performance and molecular interactions of NML-Qx-NH2 Membranes. a** Current-voltage (I−V) curves of NML-Q$_x$ and NML-Q$_x$-NH$_2$ membranes, illustrating the impact of surface functionalization on conductivity. **b** Schematic representation of the ED setup using NML-Q$_x$-NH$_2$ membranes for selective separation of ions. The ED setup consists of two CEMs on either side of a central NML-Q$_x$-NH$_2$ membrane with a membrane area of 7.07 cm². Ion transport is driven by an applied current density of 5 mA cm⁻². The electrolyte used is 0.3 mol L⁻¹ Na$_2$SO$_4$, with feed solutions of 0.1 mol L⁻¹ NaCl/Na$_2$SO$_4$, and 0.01 mol L⁻¹ KNO$_3$ (except for NO$_3$⁻ tests, where deionized water is used). and (**c, d**) Ion flux and selectivity (S) for Cl⁻ and SO$_4$²⁻ using NML-Q$_x$ and NML-Q$_x$-NH$_2$ membranes. The modified membranes show higher Cl⁻ flux and significantly enhanced Cl⁻/SO$_4$²⁻ selectivity, demonstrating the effect of surface modification on monovalent ion transport. **e** Hydration energy versus ionic radius for F⁻, Cl⁻, Br⁻, NO$_3$⁻, and SO$_4$²⁻, revealing the correlation between hydration properties and transport behavior. **f** Free energy profiles as a function of distance from the polymer center, showing preferential interaction zones for Na⁺, Cl⁻, and SO$_4$²⁻ ions. Ion and water behavior in a polymer environment from MD simulations. **g** Radial distribution functions (G(r)) for SO$_4$²⁻ with different atomic groups, highlighting differential ion-polymer interactions. **h** Radial distribution functions (G(r)) for Cl⁻ with different atomic groups in the polymer, indicating specific interactions. **i** Comparison of Cl⁻/SO$_4$²⁻ selectivity normalized to ion flux for NML-Q$_x$-NH$_2$ membranes with other reported membranes, demonstrating the high selectivity and transport efficiency achieved in this work.

hydrolysis, the ζ potential shifted to negative values, i.e., −2.33 mV to −3.05 mV. The decrease in ζ potential is attributed to -COO⁻ groups, resulting from acyl group hydrolysis, which gives an anionic character to the membranes[34]. For instance, the negative ζ potential suggest an enhanced electrostatic repulsion for SO$_4$²⁻, the high charge density of which limits its transport across the membrane. Specifically, the NML-Q$_2$-NH$_2$ membrane, with a ζ potential of −3.05 mV, exhibits strong negative charge, providing significant repulsion for SO$_4$²⁻ ions, while allowing monovalent ions. Additionally, the elevated ζ potential correlates with increased membrane resistance, offering deeper insights into how surface modification affects membrane transport properties and selectivity[22]. For example, the ζ potential of the NML-Q$_2$-NH$_2$ membrane became more negative after modification, and its resistance increased from 3.82 Ω cm² to 7.4 Ω cm². This suggests the -COOH groups neutralized the QA groups, leading to more restrictive ionic

pathways. In case of NML-Q$_2$-NH$_2$, the more negative ζ potential means high increased resistance demonstrating a stronger barrier to SO$_4$²⁻ ions[35]. However, the NML-Q$_3$-NH$_2$ membrane showed a slight decrease in ζ potential to −1.7 mV, due to the high content of QA groups. This reduction in ζ potential reflects the interaction between the QA and -COOH of the membrane after modification with TMC.

To further understand the transport behavior, we investigated their current-voltage (I−V) characteristics using the equipment as given in Fig. S12. The results are given in Fig. 4a (Fig. S13). The I−V curves displayed an Ohmic region (low current density, from 0 to approximately 0.5 V), the plateau region (current density remains nearly constant from 0.5 V to -1.3 V), and the over-limiting region (sharp increase in current density above 1.3 V)[36]. In the Ohmic region, current density increases linearly with voltage, indicating efficient ion transport with minimal resistance. The NML-Q$_2$ and NML-Q$_3$

membranes showed high LCD, at 112 mA cm$^{-2}$ and 114 mA cm$^{-2}$, respectively. The plateau region, which extends up to around 1.3 V, is characterized by nearly constant current density, reflecting capacitive behavior due to surface modifications. As the IEC increases, the membrane's capacitive nature enhances, resulting in higher resistance[37]. In the over-limiting region, current density increases sharply with voltage, often linked to water splitting and ion movement under concentration polarization. Surface modification with TMC enhances ion permselectivity by increasing $R_M$, reducing LCD, and increasing susceptibility to concentration polarization. However, the modified membranes still maintained high LCD, suggesting that the TMC-based modifications did not significantly reduce these values (Fig. S13). By correlating the ζ potential with their I-V characteristics (Fig. 4a), we observe that -COOH groups likely influences anion transport due to electrostatic repulsion effects. There are notable differences in the characteristics of membranes made using the ISIP method and that of IP approach. The conventional IP method, which employs two monomers, results in the formation of nanometer-thick surface layer, leading to a large negative ζ potential. In contrast, the ISIP technique uses a single monomer, produces membranes with improved selectivity[38]. The high ζ potential led to swelling and reduced selectivity.

## Performance evaluation and stability

To evaluate permselectivity between mono- and divalent anions, the prepared membranes were tested via ED using two types of feed systems (ED setup as given in Fig. 4b). The first was a mixture of equimolar 0.1 mol L$^{-1}$ NaCl and Na$_2$SO$_4$. The second was a mixture of 0.1 mol L$^{-1}$ NaBr, NaCl, NaNO$_3$, NaF, and Na$_2$SO$_4$. The initial phase of the study focused on Cl$^-$/SO$_4^{2-}$ as a representative system. The results of NML-Q$_x$ and its modified version, NML-Q$_x$-NH$_2$, membranes are given in Fig. 4c, d. For comparison, the base membrane (i.e., NML) was tested, which does not have IEC; however, the pyridine nitrogen can still facilitate ion exchange under the influence of electric current. The NML membrane showed a Cl$^-$ flux of 0.95 mol m$^{-2}$ h$^{-1}$, while the SO$_4^{2-}$ flux was 0.21 mol m$^{-2}$ h$^{-1}$, resulting in an ion selectivity of 4.6. As the IEC increased from 0.34 mmol g$^{-1}$ to 1.00 mmol g$^{-1}$, the ionic flux also improved. For example, compared to the base membrane, the Cl$^-$ flux increased by 22% and 4.5% for NML-Q$_2$ and NML-Q$_3$, respectively, while the SO$_4^{2-}$ flux increased by 28% and 19%, with the highest permselectivity observed at 7.7. The increases in flux align with the trend in the IEC. As the ion flux increased with higher IECs, the divalent ion flux also increased, compromising permselectivity. To mitigate the rise in divalent ion flux, the IEC was controlled at 1.00 mmol g$^{-1}$ to maintain a balanced flux and permselectivity. This trade-off can be understood through the ion transport process, where adsorption on the membrane surface is crucial in determining permselectivity[39–41]. Ion transport is significantly influenced by the Stokes radius of the ion (Table S6). For example, SO$_4^{2-}$, with a larger Stokes radius (2.31 Å), faces more resistance compared to Cl$^-$ (1.21 Å)[35]. Another factor influencing ion flux is membrane hydrophilicity. As shown in Table S7, increased swelling ratio and decreased contact angle from NML-Q$_1$ to NML-Q$_3$ reflect improved water uptake and surface wettability, which support higher Cl$^-$ transport. As previously reported by Li and Xu[42], the hydration energy of SO$_4^{2-}$ ions (ca. 1145 kJ mol$^{-1}$) is much higher than that of Cl$^-$ ions (317 kJ mol$^{-1}$), indicating that SO$_4^{2-}$ is more strongly hydrated and thus less likely to adhere to the membrane surface (Fig. 4e). Moreover, Sata et al.[43], also noted that membranes with hydrophobic surfaces have lower affinity for highly hydrated ions like SO$_4^{2-}$. As the membranes are progressively modified (from NML-Q$_1$ to NML-Q$_3$-NH$_2$), Cl$^-$ ion flux increases substantially, accompanied by a significant rise in current efficiency from 50.47% in the unmodified NML to over 110% in NML-Q$_1$-NH$_2$ (Table S8). These enhancements demonstrate that structural and chemical modifications improve the membrane's selectivity and transport efficiency for monovalent ions

like Cl$^-$. In addition, the energy consumption for Cl$^-$ separation decreases notably with membrane modification from 1.36 kWh mol$^{-1}$ in NML-Q$_1$-NH$_2$ to just 0.57 kWh mol$^{-1}$ in NML-Q$_3$-NH$_2$ further confirming the efficiency gains achieved through surface modification (Table S9).

As the IEC increases, the transport of hydrophilic ions like SO$_4^{2-}$ is more significantly enhanced compared to Cl$^-$, due to the stronger effect of IEC on ion flux, which leads to a reduction in permselectivity. Following surface modification, a significant change in ion transport behavior and permselectivity was observed. The flux of Cl$^-$ increased up to 2 mol m$^{-2}$ h$^{-1}$, whereas the flux of SO$_4^{2-}$ ions remained significantly lower. The optimized membrane (NML-Q$_2$-NH$_2$) achieved a Cl$^-$ flux of 1.98 mol m$^{-2}$ h$^{-1}$, which was higher than that of the base membrane (1.34 mol m$^{-2}$ h$^{-1}$). Meanwhile, the SO$_4^{2-}$ flux was only 0.0081 mol m$^{-2}$ h$^{-1}$, and the permselectivity was observed to be 244.

To gain further molecular-level insight, MD simulations were performed to examine the transport and interactions of ions within a representative polymer system, which mimics the membrane structure (Fig. S14, and Table S10). First, the diffusion coefficients (D values) for water, Na$^+$, Cl$^-$, and SO$_4^{2-}$ ions in the pure water system were established. Notably, SO$_4^{2-}$ showed a significantly lower diffusion coefficient compared to Cl$^-$. This aligns well with the observation that SO$_4^{2-}$ ion mobility through the membrane was lower compared to Cl$^-$, due to its larger hydrated radius, which from a radial distribution analysis was found to be 0.358 nm, compared to 0.315 nm for Cl$^-$. This is also well aligned with the stronger interactions with the membrane surface, demonstrated by the lower free energy shown in Fig. 4f. When the polymer was introduced into the system, a general reduction in the mobility of all species was observed, with the D values for Cl$^-$ and SO$_4^{2-}$ reduced to 49% and 29% of their baseline values, respectively. This hindrance was distinct for SO$_4^{2-}$, supporting the experimental finding that higher IEC levels, which increase membrane hydrophilicity, led to an increased flux of divalent ions like SO$_4^{2-}$. Free energy profiles were obtained from Boltzmann-weighted density profiles of the ions across the polymer-water phase, which reinforce the idea that larger and more hydrated ions like SO$_4^{2-}$ are more likely to interact and be retained in the polymer matrix, thus impacting their transport more significantly than smaller ions like Cl$^-$. On the other hand, Na$^+$, being a cation, experiences a higher energy barrier to enter the polymer gallery, likely due to repulsive electrostatic interactions with the QA group (Fig. 4f). The free energy profiles showed a more favorable interaction of SO$_4^{2-}$ with the polymer compared to Cl$^-$, with higher IECs. This suggests that while higher IEC enhances the transport of more hydrophilic ions like SO$_4^{2-}$, it also leads to reduced selectivity, as the membrane surface becomes more receptive to these ions. The radial distribution functions (G(r)) for Cl$^-$ with different atomic groups in the polymer (Fig. 4g) and for SO$_4^{2-}$ (Fig. 4h) clarify the differential ion-polymer interactions in the overall transport behavior. The combination of experimental results and MD simulations show that by increasing the IEC of the membranes improves the transport of hydrophilic ions like SO$_4^{2-}$, which compromise the selectivity. Hence, the ion size and hydration energy, play a critical role in governing ion transport and selectivity.

We compared our results with recent studies on monovalent anion-selective membranes (Fig. 4i). Unlike LbL and IP techniques, we controlled IEC and WU while grafting a small hydrophobic moiety with an -NH$_2$ terminal group for polymerization. After the ISIP process, the acyl group was converted to −COOH, minimizing resistance while enhancing selectivity.

Table S11 provides a detailed summary of advancements in AEM technologies, showcasing various methods and their outcomes. The ISIP approach in this study demonstrates improved performance compared to other modification techniques reported elsewhere[44]. By selecting the best data from Table S11, results of the NML-Q$_2$-NH$_2$ membrane can be compared with other modification strategies, to emphasize the effectiveness of the ISIP process. For instance, polyelectrolyte multilayer deposition was used to modify commercial AEMs. However, the reported selectivity was only 1.81, which is relatively low[45]. Similarly, the

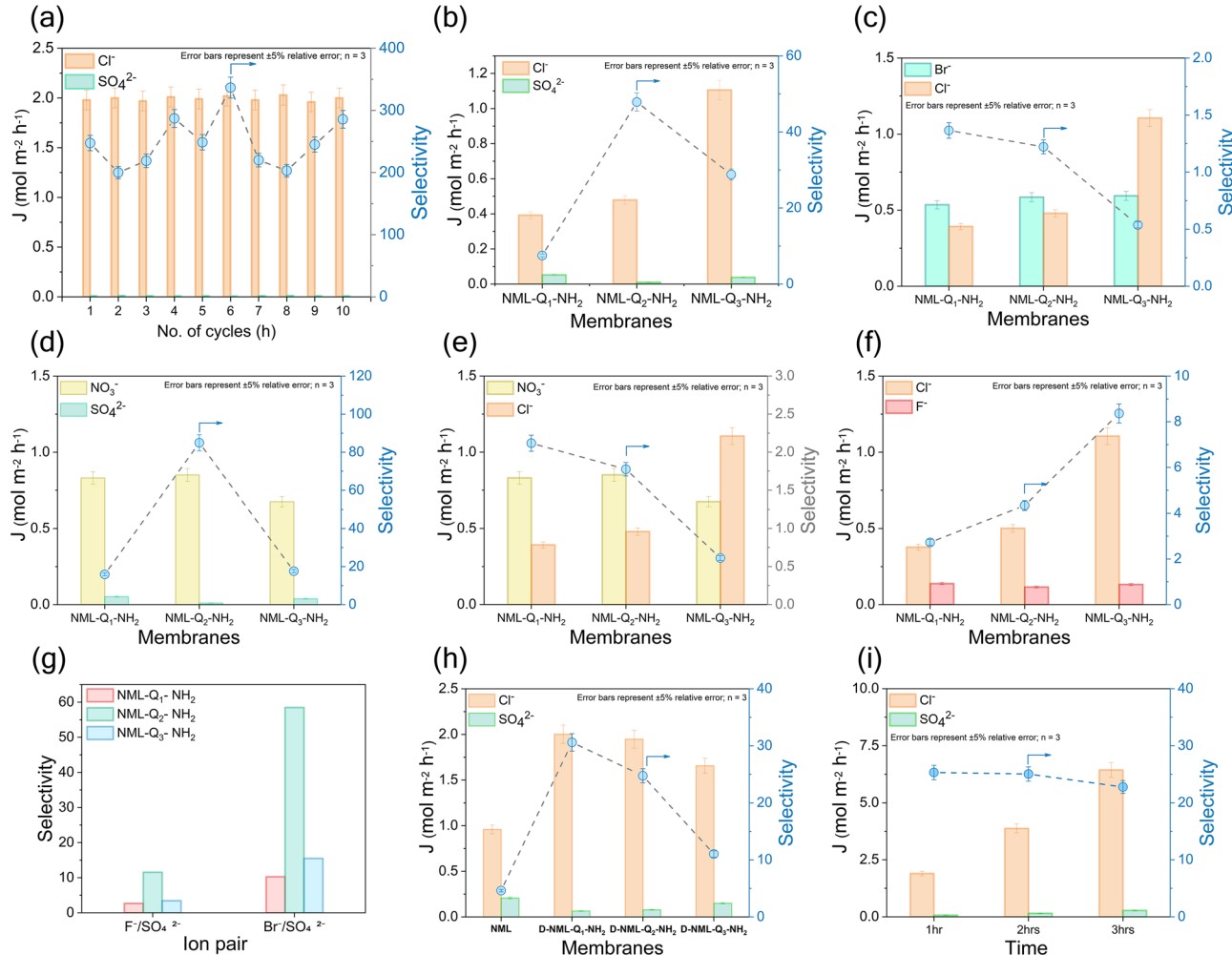

**Fig. 5 | Ion flux and selectivity characteristics of NML-Q$_x$-NH$_2$ membranes under various conditions for different ion pairs. a** Stability testing of NML-Q$_x$-NH$_2$ membranes over 10 operational cycles performed in 0.1 mol L$^{-1}$ NaCl/Na$_2$SO$_4$ solution at a current density of 5 mA cm$^{-2}$ and a membrane area of 7.07 cm$^2$. **b** Ion flux and selectivity of Cl$^-$/SO$_4^{2-}$ for NML-Q$_x$-NH$_2$ membranes. **c** Ion flux and selectivity of Br$^-$/Cl$^-$ for NML-Q$_x$-NH$_2$ membranes, showing preferential transport of Cl$^-$ and higher selectivity for the NML-Q$_3$-NH$_2$ membrane. **d** Ion flux and selectivity of NO$_3^-$/SO$_4^{2-}$ for NML-Q$_x$-NH$_2$ membranes, emphasize enhanced NO$_3^-$ selectivity due to tailored surface interaction and hydration energy differences. **e** Ion flux and selectivity of NO$_3^-$/Cl$^-$ for NML-Q$_x$-NH$_2$ membranes, illustrating improved Cl$^-$ transport with increasing surface optimization. **f** Ion flux and selectivity of Cl$^-$/F$^-$ for NML-Q$_x$-NH$_2$ membranes. **g** Comparative selectivity for F$^-$/SO$_4^{2-}$ and Br$^-$/SO$_4^{2-}$ systems, demonstrating the membrane's ability to differentiate effectively between monovalent and divalent ions. **h** The dual-modified membranes exhibit higher Cl$^-$ flux and improved selectivity. **i** ion flux and Cl$^-$/SO$_4^{2-}$ selectivity during continuous ED operation (1–3 h), indicating steady flux and sustained performance over time.

electrodeposition technique was employed to decorate the AEM with a multilayer, but the reported membranes showed a selectivity of 2.13[45]. Another study used SPPO to modify commercial AEMs, achieving a permselectivity of 52.44 and high flux. The findings are promising, but the process is difficult to scale up due to the high cost of the commercial membranes (CMXs)[46]. In contrast, our method achieves a permselectivity of 244, showing a significant improvement, especially compared to sulfonated dopamine-modified membranes, which have a permselectivity of 34.02[47].

To further evaluate the membrane's long-term performance, a stability test of the NML-Q$_2$-NH$_2$ membrane was conducted at a current density of 5 mA cm$^{-2}$, using 100 mL volumes for both feed and recovery solutions to simulate continuous operating conditions. The results proved that the membrane maintained consistent flux and permselectivity over a 10-hour period (Fig. 5a).

## Ion transport in mixed solutions and stability

The transport characteristics of various ions Cl$^-$, Br$^-$, SO$_4^{2-}$, NO$_3^-$, and F$^-$ were assessed for NML-Q$_x$-NH$_2$ membranes under a constant current density of 5 mA cm$^{-2}$ using a 0.1 mol L$^{-1}$ mixed ion solution. The experimental results, presented in Fig. 5b–g, reveal significant differences in flux across the membranes. It can be observed that the Cl$^-$ ion showed the highest flux across all three membranes. Specifically, the flux of Cl$^-$ increased significantly from NML-Q$_1$-NH$_2$ (0.39 mol m$^{-2}$ h$^{-1}$) to NML-Q$_2$-NH$_2$ (0.47 mol m$^{-2}$ h$^{-1}$), a 16.7% increase, and then surged in NML-Q$_3$-NH$_2$ (1.11 mol m$^{-2}$ h$^{-1}$), which was a 146.2% increase compared to NML-Q$_1$-NH$_2$. This large increase in Cl$^-$ flux, particularly in NML-Q$_3$-NH$_2$, was due to high IECs of the NML-Q$_3$-NH$_2$ membrane, high IECs, lower resistance and provided more channels for ion transport. On the other hand, the flux of other ions showed some mixed trend; however, in all cases the flux of SO$_4^{2-}$ was lower because of the divalent trend, high hydration energy as discussed in the above section. The flux of SO$_4^{2-}$ in NML-Q$_2$-NH$_2$ was substantially lower (0.01 mol m$^{-2}$ h$^{-1}$), an 80% decrease from NML-Q$_1$-NH$_2$, where the flux was 0.05 mol m$^{-2}$ h$^{-1}$. This drastic decrease can be attributed to the larger hydrated radius due to high charge density, which makes it less mobile. This suggests that the membrane's ability to restrict the active transport of SO$_4^{2-}$ ions is significantly enhanced. The flux of Br$^-$ increased by 9.3% from NML-Q$_1$-NH$_2$ to NML-Q$_2$-NH$_2$, from 0.54 mol m$^{-2}$ h$^{-1}$ to 0.59 mol m$^{-2}$ h$^{-1}$. However, no further increase was observed in NML-Q$_3$-NH$_2$, where the flux

remained at 0.59 mol m$^{-2}$ h$^{-1}$, suggesting that Br$^-$ transport was not significantly influenced by further changes in the membrane properties (Fig. 5c). For NO$_3^-$, the ion flux showed a slight increase from 0.83 mol m$^{-2}$ h$^{-1}$ to 0.86 mol m$^{-2}$ h$^{-1}$ i.e., a 3.6% increase, but then decreased in NML-Q$_3$-NH$_2$ to 0.68 mol m$^{-2}$ h$^{-1}$, representing an 18.4% decrease from the initial value (Fig. 5d, e). We also investigated the F$^-$, which is smaller in size but has a higher charge density than other anions, but less than SO$_4^{2-}$ and therefore showed lower ion flux (Fig. 5f, g). For instance, the NML-Q$_1$-NH$_2$ membrane showed a value of 0.14 mol m$^{-2}$ h$^{-1}$ and this decreased to 0.12 mol m$^{-2}$ h$^{-1}$. However, a small regain of 7.1% in NML-Q$_3$-NH$_2$ (with a flux of 0.13 mol m$^{-2}$ h$^{-1}$) suggests the hydrophilic behavior also affects the flux of the F$^-$ ion.

As discussed, the selectivity ratios of Cl$^-$ relative to Br$^-$, SO$_4^{2-}$, NO$_3^-$, and F$^-$, as well as the selectivity ratios of NO$_3^-$, F$^-$, and Br$^-$ relative to SO$_4^{2-}$ for the NML-Q$_1$-NH$_2$, NML-Q$_2$-NH$_2$, and NML-Q$_3$-NH$_2$ membranes, reveal important trends in ion separation. For the selectivity of Cl$^-$ relative to other ions, the NML-Q$_2$-NH$_2$ membrane generally exhibits the highest selectivity. The NML-Q$_3$-NH$_2$ membrane exhibits high permeation for Cl$^-$ ions compared to F$^-$ and SO$_4^{2-}$ ions, with Cl$^-$/F$^-$ and Cl$^-$/SO$_4^{2-}$ ratios of 8.37 and 28.79, respectively. However, its selectivity is lower than that of the NML-Q$_2$-NH$_2$ membrane, which has a selectivity ratio of 47. Looking at the selectivity of NO$_3^-$, F$^-$, and Br$^-$ relative to SO$_4^{2-}$, it shows that the NML-Q$_2$-NH$_2$ membrane has the highest selectivity for NO$_3^-$ over SO$_4^{2-}$, with a ratio of 84.99, significantly higher than NML-Q$_1$-NH$_2$ (15.92) and NML-Q$_3$-NH$_2$ (17.62). This indicates that the NML-Q$_2$-NH$_2$ membrane is more effective at rejecting SO$_4^{2-}$ ions while allowing NO$_3^-$ ions to pass through. For F$^-$/SO$_4^{2-}$ selectivity, NML-Q$_2$-NH$_2$ also exhibits the highest ratio (11.55), indicating its high selectivity (Fig. 5g).

It is clear that the NML-Q$_2$-NH$_2$ membrane favors smaller, less hydrated anions such as Cl$^-$, Br$^-$, and NO$_3^-$ over more strongly hydrated ions like SO$_4^{2-}$. Although F$^-$ is smaller in size, its moderate hydration energy (-580 kJ mol$^{-1}$) still contributes to lower permeability compared to other monovalent anions. This selectivity transport is likely driven by the membrane's surface properties such as zeta potential, hydrophilicity, and in-situ production of -COOH groups, which demonstrate a significant increase in Cl$^-$ transport and the reduction in SO$_4^{2-}$ flux.

The performance of membranes with both-sided modifications was also evaluated to enhance anion selectivity. The results reveal a significant trend in both ion flux and selectivity that are strongly influenced by the degree of crosslinking introduced during the modification process as given in Fig. 5h. A notable decrease in Cl$^-$ flux and a corresponding increase in SO$_4^{2-}$ flux (due to increase hydrophilicity of -COOH) were observed, which ultimately led to a sharp decrease in the overall selectivity of the membranes. The Cl$^-$ flux for the dual side modified (D-NML-Q$_1$-NH$_2$) membrane was measured at 2.0 mol m$^{-2}$ h$^{-1}$, which then decreased to 1.6 mol m$^{-2}$ h$^{-1}$ in D-NML-Q$_3$-NH$_2$. This represents a 17% decrease in Cl$^-$ flux, which is likely due to the dense membrane matrix on both sides. Conversely, the SO$_4^{2-}$ flux exhibited an increase from 0.06 mol m$^{-2}$ h$^{-1}$ in D-NML-Q$_1$-NH$_2$ to 0.15 mol m$^{-2}$ h$^{-1}$ in D-NML-Q$_3$-NH$_2$, a 129% increase. This increase in SO$_4^{2-}$ indicates that the membrane's selective rejection is more dependent on the specific interactions between the membrane matrix and the ions, rather than simply on the size or charge of the ions. For instance, the selectivity for Cl$^-$ flux over SO$_4^{2-}$ was 30.6 in D-NML-Q$_1$-NH$_2$, but this value dropped significantly to 11.0 in D-NML-Q$_3$-NH$_2$, representing a 64.0% decrease.

In Fig. 5i, the 1-hour value (1.90 mol m$^{-2}$) represents the amount of Cl$^-$ transported during the first hour, while the 2-hour (3.87 mol m$^{-2}$) and 3-hour (6.44 mol m$^{-2}$) values reflect the transport over two and three hours, respectively. Since these are cumulative values, an increasing trend over time is expected. Importantly, the calculated flux rates (1.90–6.44 mol m$^{-2}$ h$^{-1}$) remain relatively steady, indicating stable transport behavior. Additionally, the SO$_4^{2-}$ transport remains minimal

throughout the test (0.075–0.28 mol m$^{-2}$), and the Cl$^-$/SO$_4^{2-}$ selectivity remains high (25.31 to 22.77), demonstrating consistent separation performance during the entire operation.

Additionally, we conducted various TMC concentrations to further explore its impact on membrane performance. These findings are presented in Fig. S15. The data show that ion selectivity was very high at 0.1 wt%, but as the concentration increased to 0.15 wt% and 0.2 wt%, both selectivity and ion flux declined. This is attributed to highly crosslinked surfaces and charge accumulation, which may hinder ion transport. These findings suggest that 0.1 wt% TMC is the optimal concentration; however, further detailed investigations are necessary to validate this conclusion.

## Insights on ion transport via DFT calculations

DFT calculations were performed on a protonated and deprotonated discrete model molecule to derive atomic partial charges, which were critical for the molecular dynamics (MD) simulations, and electrostatic potential (ESP) charge analysis, as in Fig. 6a, b (details of the method are provided in Note S2). The ESP heat maps reveal a significant difference in charge distribution between the protonated (-COOH) and deprotonated (-COO$^-$) states of the model molecule, providing valuable insights into the ion transport mechanisms governing selective permeability. In the unmodified membrane, positive charges are primarily localized on the pyridinium units (QA), facilitating the transport of counterions (Fig. 6a). Following deprotonation, charge redistribution occurs, which could lead to partial delocalization of the negative charge over the large area of the model molecule, which contains anions with varying charge densities and hydration sizes. This distribution of negative charge generates electrostatic repulsion, particularly for larger divalent anions (Fig. 6b). As a result, the transport of these anions is hindered. On the other hand, smaller anions like Cl$^-$, with lower charge density and a smaller hydration shell, experience less repulsion and can move more freely. Another key finding from the DFT calculations is that the deprotonation of the linker group and these structural changes narrowed the pathways on the surface. If translated to the membrane, such a modification affects the size and accessibility of ion transport channels. Consequently, the degree of deprotonation not only influences the electrostatic environment but, by extension, also affects the membrane's overall morphology, which plays a pivotal role in ion selectivity.

## Discussion

In conclusion, the development of MAPMs using the ISIP technique marks a significant advancement in selective ion separation technologies. This study demonstrated the successful fabrication and functionalization of MAPMs by introducing amine (-NH$_2$) and cationic groups into the membrane structure, forming a crosslinked polyamide structure with TMC. The modified membranes exhibit a dense, negatively charged surface that enhances their permselectivity, as confirmed by zeta potential measurements and ATR-FTIR analyzes. Specifically, the ATR-FTIR spectra revealed the formation of characteristic amide linkages, aromatic frameworks, and carboxylic acid (-COOH) groups, all of which contribute to the observed improvements in membrane performance. The performance evaluation highlighted the remarkable properties of these membranes, such as a high limiting current density of over 90 mA cm$^{-2}$, a low membrane resistance of 4.7 Ω cm², and a substantial Cl$^-$ flux of 2.1 mol m$^{-2}$ h$^{-1}$. These findings, combined with a high selectivity of approximately 244 for Cl$^-$/SO$_4^{2-}$, emphasize the effectiveness of the membranes in discriminating against larger, more hydrated ions like SO$_4^{2-}$ while favoring smaller ions such as Cl$^-$. Selectivity for other anions, including Br$^-$, F$^-$, and NO$_3^-$, was also demonstrated, further validating the robustness of the ISIP process for achieving targeted ion separation. The NML-Q$_x$-NH$_2$ membranes consistently showed higher selectivity for Cl$^-$ and Br$^-$,

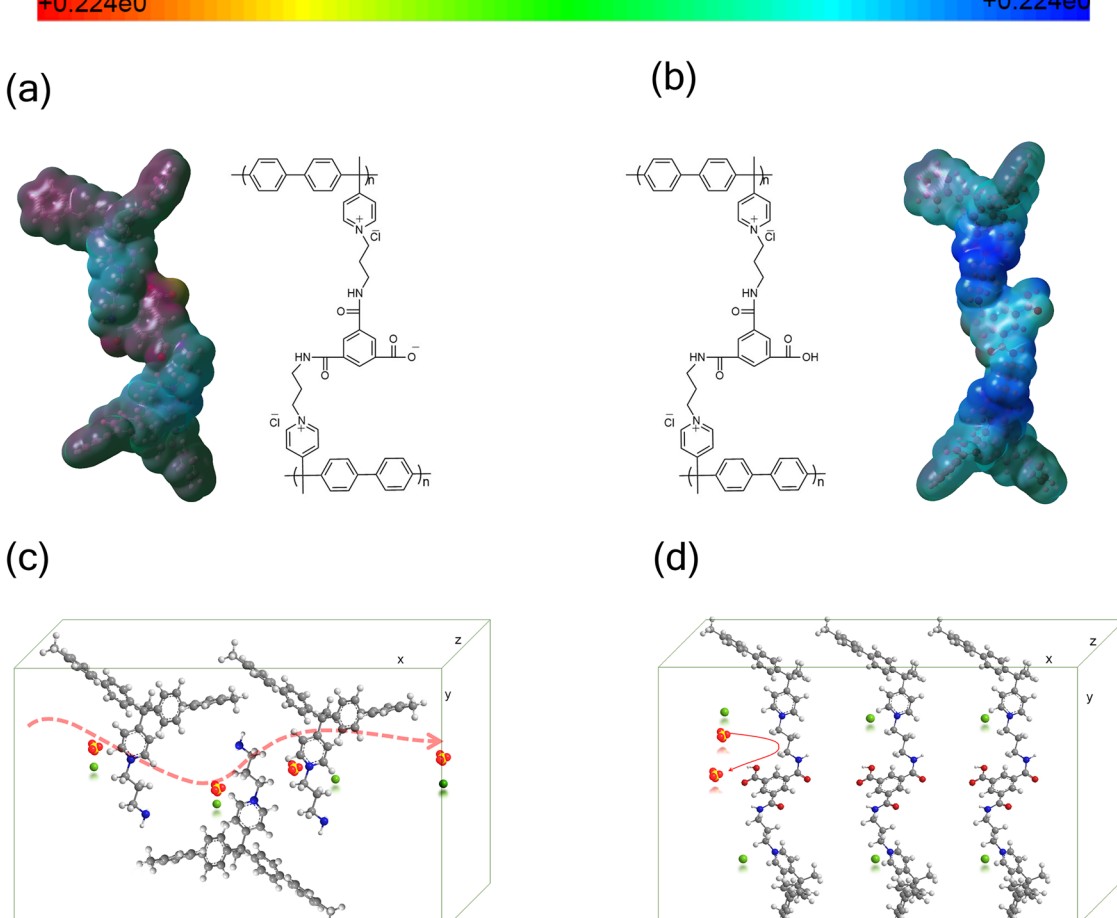

**Fig. 6 | Structural and functional insights into crosslinked polymer membranes. a** Crosslinked deprotonated polymer with bulky -COO⁻ groups near the surface, enhancing Cl⁻ permeability and repelling SO₄²⁻ through electrostatic effects. **b** Protonated crosslinked polymer adopting a compact conformation due to reduced electrostatic repulsion (heat map range: +0.224e₀ to −F0.224e₀). **c** Base (non-crosslinked) polymer showing minimal anion selectivity during ion transport. **d** Crosslinked polymer enabling selective transport Cl⁻ passes freely, while SO₄²⁻ is effectively rejected, demonstrating improved monovalent ion selectivity.

driven by surface characteristics such as increased hydrophilicity, negative zeta potential, and the functional presence of -COOH groups. These results establish ISIP-based MAPMs as a highly promising solution for efficient and selective ion separation, with potential applications in water desalination, energy storage systems, and industrial separation processes. The scalability and precision of the ISIP method offer a pathway to optimizing membrane properties for specific ions, paving the way for further research and innovations in advanced membrane technology.

## Methods

### Materials
All chemicals used were of analytical grade. Biphenyl, 4-acetylpyridine, trifluoromethanesulfonic acid (TFSA), and trifluoroacetic acid (TFA) were obtained from Saan Chemical Technology Co., Ltd., in Shanghai, China. 3-Bromopropan-1-amine was supplied by Shanghai Aladdin Biochemical Technology Co., Ltd., also in Shanghai, China. Sodium Cl⁻ (NaCl), sodium fluoride (NaF), sodium bromide (NaBr), sodium nitrate (NaNO₃), sodium sulfate (Na₂SO₄), potassium nitrate (KNO₃), N-methyl-2-pyrrolidone (NMP), dichloromethane (DCM), trimesoyl chloride (TMC), and dimethyl sulfoxide (DMSO) were purchased from Sinopharm Group Chemical Reagent Co., Ltd., based in Beijing, China. The cation exchange membrane (CEM) and monovalent anion perm-selective membrane (ACS) were acquired from ASTOM, located in Tokuyama Soda, Japan.

### Preparation of poly (alkyl-biphenyl pyridine) (PAB)
To synthesize PAB, biphenyl (6 g, 38.9 mmol) and 4-acetylpyridine (6.3 g, 52.06 mmol) were combined in a flask in a molar ratio of 1:1.34, along with dichloromethane (10 mL), and mechanically stirred for 20 mins. Under ice bath conditions, 2 mL of trifluoroacetic acid (TFA) and 22 mL of trifluoromethanesulfonic acid (TFSA) were slowly added to the reaction mixture. The mixture was stirred at 0 °C for 2 h, then allowed to stir at room temperature for 24 h. During this period, the solution gradually developed a purplish-red color. The resulting purplish-red solution was carefully poured into deionized water, leading to the formation of a yellow-fibrous precipitate. The product was stirred in a 1 mol L⁻¹ NaOH solution for 24 h to neutralize any residual acid and ensure complete deprotonation of the polymer. Afterward, the polymer was filtered and thoroughly washed with deionized water to remove any remaining impurities. The deprotonated (PAB) was then dried at 40 °C for 24 h.

### Quaternization of PAB polymer
For the quaternization, PAB (2.15 mmol) was first dissolved in 9 mL of DMSO at 25 °C. Various amounts of amine, as shown in Table S1, were added to control the IEC values. The reaction was carried out at 70 °C for 12 h. The membrane was prepared using the solution casting method. After the quaternization reaction, the polymer solution was filtered through a 0.5 μm syringe filter and cast onto a clean glass plate,

with the edges sealed using ~0.1 mm thick transparent insulation tape to prevent outflow. The cast film was dried at 60 °C for 6–7 h to fully evaporate the solvent. The dried membrane was then gently detached by soaking in water, thoroughly washed, and converted to the Cl⁻ form by immersing it in a 0.5 mol L⁻¹ NaCl solution for 24 h.

For comparison, a reference membrane (NML) was prepared using the same method without amine addition, meaning no quaternization occurred.

### In situ surface-crosslinking
Each membrane in Cl⁻ form (10 × 12 cm) was securely held in a specially designed holder to ensure surface modification occurred on only one side. The holder was engineered to maintain a firm grip on the membrane. To initiate surface modification, various solutions of TMC 0.1 wt%, 0.15 wt%, and 0.25 wt% were poured onto the membrane surface. After 180 seconds, the membranes were washed three times with n-hexane to remove any unreacted TMC. The membranes were then placed in an oven and dried at 60 °C for 5 min. After drying, the membranes were hydrated to convert any remaining unreacted acyl groups into -COOH groups. To further assess the ISIP process and surface tuning effects on ion selectivity, both sides of the membranes were modified. A membrane piece was immersed in TMC solution for 3 min, then dipped in n-hexane 2–3 times to remove unreacted TMC. The hydrated membranes were then characterized and tested for application.

### Membrane characterization
The surface characteristics of the NML-Q$_x$ membranes were comprehensively analyzed and compared to the base membranes using multiple techniques. X-ray photoelectron spectroscopy (XPS) with an Al Kα X-ray source (Thermo ESCALAB250, USA) and atomic force microscopy (AFM, Multi-Mode V, USA) provided insights into their surface composition and topography. The presence of functional groups such as amide linkages and -COOH groups was confirmed through attenuated total reflectance–Fourier transform infrared (ATR-FTIR) spectroscopy (Vector 22, Bruker) over the spectral range of 4000–650 cm⁻¹. Additionally, zeta potential measurements, performed using the SurPASS electrokinetic instrument (Anton Paar Trading Co., China), offered valuable information on the membranes' surface charge characteristics[48]. Static water contact angles were measured at room temperature using an Ossila Goniometer with a 0.47 mm blunt needle. Values are reported as mean ± standard deviation from multiple measurements.

### Ion exchange capacity
The ion exchange capacity (IEC) of the membranes was measured using Mohr's method. Dry membranes were accurately weighed and converted to the Cl⁻ form by immersing them in a 1.0 mol L⁻¹ NaCl solution for 24 h. Subsequently, the membranes were thoroughly rinsed with deionized water to remove excess salt and immersed in a 0.5 mol L⁻¹ Na₂SO₄ solution for 48 h to release Cl⁻ ions. The released Cl⁻ ions were titrated with a 0.01 mol L⁻¹ AgNO₃ solution, using K₂CrO₄ as the indicator. The IEC (mmol g⁻¹) was then calculated using Eq. 1.

$$IEC = \frac{\left(V_{AgNO_3} \times C_{AgNO_3}\right)}{W_{dry}} \quad (1)$$

where $W_{dry}$ (g) is the weight of the dried membrane, $V$ (mL) is the volume of the AgNO₃ solution, and $C$ (mol L⁻¹) is the molar concentration of the AgNO₃ solution.

### WU and swelling ratio (SR)
The WU was measured by soaking the membranes in water for 24 h and then weighing them as $W_2$. The membranes were then dried at 60 °C

for 12 h and re-weighed as $W_1$ (dried membrane). The WU of the membranes was calculated using the following Eq. 2:

$$WU = \frac{(W_2 - W_1)}{W_1} \times 100 \quad (2)$$

$W_1$ and $W_2$ represent the weight of the dried and wet membrane, respectively.

The swelling ratio (SR) was calculated using Eq. 3 as the area difference between wet ($S_{wet}$, cm²) and dry ($S_{dry}$, cm²) membrane samples.

$$SR(\%) = \frac{S_{wet} - S_{dry}}{S_{dry}} \times 100 \quad (3)$$

### Transport number ($t_{Cl^-}$)
The transport number ($t_{Cl^-}$) of the membranes was measured using a two-compartment cell. The membrane sample was held between two compartments containing NaCl solutions of different concentrations, i.e., 0.01 mol L⁻¹ and 0.05 mol L⁻¹, respectively. The potential difference generated across the membrane was observed with a multimeter attached to two Ag-AgCl electrodes. The $t_{Cl^-}$ of Cl⁻ was calculated using the Eq. (4) (Nernst equation) as follows:

$$E_m = \frac{RT}{zF}(2t_i - 1)\ln\frac{a_1}{a_2} \quad (4)$$

$E_m$ (in volts) is the membrane's potential, R is the universal gas constant, $T$ is the absolute temperature (K), $F$ (Faraday constant), and $z$ (ion charge = 1 for Cl⁻); whereas $a_1$ and $a_2$ are the activities of the electrolyte in the solutions, respectively.

### Current-voltage (I–V)
The current-voltage (I–V) curves were obtained by a four-compartment apparatus using 0.5 mol L⁻¹ NaCl and 0.3 mol L⁻¹ Na₂SO₄ solutions, respectively. The current was increased progressively using a potentiostat/galvanostat (WYL1703, Hangzhou Siling Electrical Instrument Ltd.) and the voltage drop across a membrane (with an effective area of 3.2 cm²) was observed with a multimeter attached with a pair of Ag-AgCl electrodes.

### Anion perm selectivity
The membrane's anion permselectivity was measured using a lab-made ED setup in which the membrane sample was seized in the center of the ED stacks and the modified surface was facing the diluted chamber. CMX were clamped on either side of the modified membrane to allow the electromigration of cation to fulfill the principle of electroneutrality. The diluted compartment was filled with 100 mL solution containing equimolar (0.1 mol L⁻¹) of NaCl and Na₂SO₄, while the concentrated compartment was filled with a 100 mL of 0.01 mol L⁻¹ KNO₃ solution, respectively. In the electrolyte compartment, 100 mL of 0.3 mol L⁻¹ Na₂SO₄ solution was used. Each solution was pumped continuously via peristaltic pumps with a flow rate of 20 mL min⁻¹ to avoid concentration polarization. A constant current density of 5 mA cm⁻² was used to initiate the electromigration of ions across the membrane. After 1 h running time, the concentration of Cl⁻ was tested by titrating with 0.01 mol L⁻¹ AgNO₃, while the SO₄²⁻ was tested using the ICP analysis (ICP-AES, Optima 7300 DV, USA). The flux J$_i$ (mol m⁻² h⁻¹) of the anions was calculated using the following Eq. (5):

$$J_i = \frac{(C_t - C_i) \cdot V}{A_m \cdot t} \quad (5)$$

$V$ is the volume (0.1 L), $C_t$ and $C_i$ represent the final and initial molar concentration (mol L$^{-1}$) in the concentrated compartment, $t$ is the time (hour), $A_m$ is the effective area (m$^2$) of membrane. The perm selectivity of the membrane was calculated using the following Eq. (6):

$$P^{Cl^-}_{SO_4^{2-}} = \frac{t_{Cl^-}/t_{SO_4^{2-}}}{C_{Cl^-}/C_{SO_4^{2-}}} = \frac{J_{Cl^-}}{J_{SO_4^{2-}}} \qquad (6)$$

$C_{Cl^-}$ and $C_{SO_4^{2-}}$ are the molar concentration of NaCl and Na$_2$SO$_4$, respectively (0.1 mol L$^{-1}$). Whereas, $t_{Cl^-}$ and $t_{SO_4^{2-}}$ are the transport numbers of Cl$^-$ and SO$_4^{2-}$, respectively.

The current efficiency was calculated using Eq. (7)[49]:

$$\text{Current Efficiency } (\eta\%) = \left(\frac{z_i F V \Delta C_i}{It}\right) \times 100 \qquad (7)$$

where ($\Delta C_i$) is the change in ion concentration (mol L$^{-1}$), $z_i$ is the ion charge, $F$ is Faraday's constant (C mol$^{-1}$), $V$ is the volume of solution (L), $I$ is the current (A), and $t$ is time (s). Moreover, energy consumption ($E$) per mole of ion was calculate for according to the Eq. (8)[50]:

$$\text{Energy consumption } (E) \text{ per mole of ion} = \frac{\int V.I.dt}{(C_i - C_t).V} \qquad (8)$$

where $V$ is the voltage drop across the membrane (we assumed 12.5 V), $I$ is the applied current (0.036 A), $dt$ (s) is the time change, and $C_i$ and $C_t$ are the concentrations of ion initially and at a specific time $t$ (s).

## Data availability
All data supporting the findings of this study are available in the article and the Supplementary Information. Source data is provided as a Source Data file. Source data are provided with this paper.

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

## Acknowledgements

We acknowledge financial support from the National Natural Science Foundation of China (grant numbers 22222812 and 22178330), the Kempe Foundation (grant number JCK22-0008), and the Swedish Research Council (grant number 2023-04608). We also thank Sourov Mondal for his assistance with the experimental work.

## Author contributions

N.U.A. conceived the research, conducted the experiments, wrote the original manuscript, and revised the manuscript. M.H. performed molecular dynamics simulations, contributed to the initial draft, and participated in manuscript revision. A.O. carried out the DFT calculations, contributed to the initial draft, and revised the manuscript. N.A.K. contributed to revising the first draft. G.L. supervised the project, acquired funding, contributed to the original draft, and revised the manuscript. T.X. provided comments on both the original and revised versions of the manuscript. N.T. conceived the research, supervised the project, acquired funding, contributed to the original draft, and revised the manuscript.

## Funding

## Competing interests

The authors declare no competing interests.
