## [Peer Review file · Nature Communications]

Monovalent anion-selective membranes fabricated via in-situ interfacial polymerization

Corresponding Author: Professor Naser Tavajohi

Version 0:

Reviewer comments:

Reviewer #1

(Remarks to the Author)

The selective separation of anions is a highly challenging area of research. The authors have ingeniously developed a straightforward ISIP method to design monovalent anion-selective membranes. By incorporating a negatively charged cross-linked layer on the surface of the ion-exchange membrane, they have successfully enhanced the membrane's anion selectivity and ion flux. While various innovative materials, such as COFs and MOFs, have been utilized to construct ion-selective membranes with excellent separation performance, the approach proposed by the authors remains highly significant for practical applications in polymer-based ion-selective membranes. In principle, the manuscript can be accepted; however, there are several details that could benefit from further attention and refinement. Below are specific suggestions:

1. It would be beneficial if the authors carefully reviewed and improved the figures in the manuscript, paying particular attention to color coordination and layout design.
2. Fluoride ions should be classified as weakly hydrated ions rather than strongly hydrated ones like sulfate ions. Clarifying the Gibbs hydration free energy of fluoride ions would help avoid potential misunderstandings regarding the membrane's separation performance.
3. In the abstract, it would be helpful to include both the current density and the membrane's ion flux and selectivity values. Additionally, there appears to be a discrepancy between the ion flux value mentioned in the abstract (2.1) and the one in the main text (1.98 at line 335).
4. At line 103, the term "permselective anion-selective membranes" is used; it might be advantageous to adopt consistent terminology throughout the manuscript.
5. In Figure 1, the accompanying ion of the quaternized product NML-Qx, generated by the reaction between polymer NML and amine, should be bromide rather than chloride. Additionally, since the authors employed the ISIP method, which indicates that the reaction occurs only on the membrane surface, the chemical reaction expression in Figure 1 might lead to confusion. We kindly suggest revising it for clarity.
6. At line 142, should it refer to Figure 3f instead of Figure 2f?
7. Some expressions appear somewhat lengthy. It would be beneficial to streamline them for better readability. For example, the sections introducing the three-phase membrane structure and describing phase separation could be simplified.
8. The authors are encouraged to clearly specify the conditions of electro dialysis in the figure captions to avoid any potential confusion.
9. How does the AFM image in Figure 2b reflect the microphase separation structure of the membrane? Given that the tested area corresponds to the membrane's surface layer (i.e., the ISIP layer), further clarification is recommended.
10. Figure 4i presents the results of a continuous electro dialysis process. Could the authors elaborate on how these results support the viewpoint mentioned at line 310?
11. Overall, the membranes described in the manuscript exhibit high ion fluxes. It is suggested that the authors calculate the current efficiency to demonstrate that such high ion fluxes fall within a reasonable range.
12. The authors are kindly advised to carefully review the description of ion flux values from lines 450 to 457. Did the authors calculate the ion flux without considering the time factor? Ideally, the ion flux should remain relatively stable over time rather than increasing significantly.
13. In the experimental section, DPAB is used to represent the deprotonated polymer, whereas PAB is utilized in the main text. A proofreading check is recommended to ensure consistency.

Reviewer #2

(Remarks to the Author)

[Editor Note: See Attached Document]

Reviewer #3

(Remarks to the Author)

In this work, MD simulations were performed to examine the transport and interactions of ions within the polymer system. The authors need further clarification on the following questions.

1. In the MD simulation methods, what is the meaning of 'a small excess of the resp.'
2. How is the electric field set during the simulation?
3. Is the simulation system maintaining neutral? How is the neutrality maintained?
4. The exact procedure for calculating the free energy profiles needs to be given in detail. How are the Boltzmann-weighted density profiles obtained?
5. 'due to its larger hydrated radius and stronger interactions with the membrane surface (hydrophilicity)', please support the above conclusion with data from molecular simulations.

Version 1:

Reviewer comments:

Reviewer #1

(Remarks to the Author)

As all my concerns are well-resolved, the manuscript can be accepted for publication.

Reviewer #2

(Remarks to the Author)

Thank you for the thorough and thoughtful revision. The authors have meticulously addressed all previous concerns, significantly improving the manuscript. The revised manuscript demonstrates significant improvements in clarity, experimental detail, and data interpretation.

The work presents a well-executed approach to fabricating monovalent anion-selective membranes via in situ interfacial polymerization. The reported $\text{Cl}^-/\text{SO}_4^{2-}$ selectivity (up to 244), along with low membrane resistance and high ion flux, represents a meaningful advancement over existing membrane technologies. The inclusion of simulation data further strengthens the mechanistic understanding.

Methods are described in sufficient detail for reproducibility, and the conclusions are well supported by the data. This study will be of interest to researchers working in membrane science, ion separations, and water treatment.

I recommend acceptance in its current form.

Reviewer #3

(Remarks to the Author)

The authors have comprehensively responded our comments regarding the simulation details. It can be accepted as it is.

Responses to reviewers' comments

We sincerely thank reviewers for their thoughtful comments and constructive suggestions. In response, we have carefully revised the main manuscript and the Supplementary Information (SI) file. All figures, figure numbers, and references have been double-checked and corrected where necessary. To ensure clarity, all changes made to the manuscript and SI are highlighted in red. In the response letter, our replies are provided in black, Times New Roman font, and any newly inserted text is shown in italics. Below, we provide a point-by-point response to each of the reviewers' comments.

Reviewer #1 (Remarks to the Author):

General comment: The selective separation of anions is a highly challenging area of research. The authors have ingeniously developed a straightforward ISIP method to design monovalent anion-selective membranes. By incorporating a negatively charged cross-linked layer on the surface of the ion-exchange membrane, they have successfully enhanced the membrane's anion selectivity and ion flux. While various innovative materials, such as COFs and MOFs, have been utilized to construct ion-selective membranes with excellent separation performance, the approach proposed by the authors remains highly significant for practical applications in polymer-based ion-selective membranes. In principle, the manuscript can be accepted; however, there are several details that could benefit from further attention and refinement. Below are specific suggestions:

Response: Thank you very much for your encouraging comments. We truly appreciate that you found our approach to be both innovative and practically relevant. We agree that materials such as COFs and MOFs have demonstrated great performance for ion separation. However, they often present challenges including high cost, complex synthesis procedures, and limited long-term stability, which can hinder their applicability in real-world scenarios. Our ISIP-based method, by contrast, employs a polymer system that is easier to process, more stable, and readily scalable. This makes it better suited for practical uses, such as water treatment and resource recovery. We believe this is one of the main advantages of our work.

We carefully considered all your detailed suggestions and made improvements to the manuscript to make it clearer. Our point-by-point responses to each suggestion are provided below.

Q 1. It would be beneficial if the authors carefully reviewed and improved the figures in the manuscript, paying particular attention to color coordination and layout design.

Response: Thank you for your suggestion. We carefully reviewed all the figures in both the main manuscript and the Supporting Information. We have adjusted the color schemes to improve clarity. Additionally, we have improved the figures layout to make them easier to interpret and more visually organized.

Q 2. Fluoride ions should be classified as weakly hydrated ions rather than strongly hydrated ones like sulfate ions. Clarifying the Gibbs hydration free energy of fluoride ions would help avoid potential misunderstandings regarding the membrane's separation performance.

Response: Thank you for your insightful comment. We agree with your observation. F^- is not as strongly hydrated as SO_4^{2-} and should be classified as a weakly hydrated ion. Therefore, we have rephrased the sentence in the revised manuscript as follows (see lines 413-416):

“It is clear that the NML-Q₂-NH₂ membrane favors smaller, less hydrated anions such as Cl^- , Br^- , and NO_3^- over more strongly hydrated ions like SO_4^{2-} . Although F^- is smaller in size, its moderate hydration energy (-580 kJ mol^{-1}) contributes to its lower permeability compared to other monovalent anions.”

Q 3. In the abstract, it would be helpful to include both the current density and the membrane's ion flux and selectivity values. Additionally, there appears to be a discrepancy between the ion flux value mentioned in the abstract (2.1) and the one in the main text (1.98 at line 335).

Response: Thank you for your comment. The Cl^- ion flux has been corrected to match the value reported in the main text, which is $1.98 \text{ mol m}^{-2} \text{ h}^{-1}$. Additionally, we have updated the abstract to include the current density (5 mA cm^{-2}), Cl^- flux ($1.98 \text{ mol m}^{-2} \text{ h}^{-1}$), and Cl^-/SO_4^{2-} selectivity (244). According to the journal's author guidelines, the abstract should be limited to 150 words or fewer. Therefore, we have revised the abstract as follows:

Updated sentence (lines 13–25):

“Developing monovalent anion-selective membranes (MAPMs) faces challenges, including the trade-off between flux and selectivity, membrane stability, and cost-effective fabrication. Overcoming these requires advanced material design and scalable techniques. Here, we introduce in situ interfacial polymerization (ISIP) to prepare MAPMs. Base membranes were synthesized via superacid polymerization and modified with anion channels and $-NH_2$ groups. During ISIP, trimesoyl chloride reacted with surface $-NH_2$ groups, forming a partially crosslinked structure with $-COOH$ groups to regulate ion transport via electrostatic interactions. This led to low membrane resistance ($4.7 \Omega cm^2$) and selective transport of weakly hydrated ions (Cl^- , Br^- , NO_3^-), while strongly hydrated ions (SO_4^{2-} , F^-) faced higher barriers. MAPMs demonstrated high performance, achieving a limiting current density ($>90 mA cm^{-2}$), Cl^- flux ($1.98 mol m^{-2} \cdot h^{-1}$ at $5 mA cm^{-2}$), and selectivity (244 for Cl^-/SO_4^{2-}), confirming effective hydration dynamics control and balanced performance. Simulations revealed how charge distribution affects ion migration pathways.”

Q 4. At line 103, the term "permselective anion-selective membranes" is used; it might be advantageous to adopt consistent terminology throughout the manuscript.

Response: Thank you for your constructive suggestion. We have revised the terminology throughout the manuscript, using “anion permselective membrane” instead of “permselective anion exchange membrane,” to uniformity. Please see line 527. *“The cation exchange membrane (CEM) and monovalent anion permselective membrane (ACS) were acquired from ASTOM, located in Tokuyama Soda, Japan.”*

Q 5. In Figure 1, the accompanying ion of the quaternized product NML-Qx, generated by the reaction between polymer NML and amine, should be bromide rather than chloride. Additionally, since the authors employed the ISIP method, which indicates that the reaction occurs only on the membrane surface, the chemical reaction expression in Figure 1 might lead to confusion. We kindly suggest revising it for clarity.

Response: Thank you for your comment. We acknowledge that the quaternized NML-Qx membranes initially have Br^- as the counterions. To clarify, prior to the TMC surface modification, the membranes were thoroughly washed and converted to the Cl^- form by soaking them in $0.5 mol L^{-1}$ NaCl solution. This step has now been explicitly added to the Experimental section to avoid confusion. Specifically, please see the statement under the heading **Quaternization of PAB polymer** at line 550-552: *“The dried membrane was then gently*

detached by soaking in water, thoroughly washed, and converted to the Cl⁻ form by immersing it in a 0.5 mol L⁻¹ NaCl solution for 24 hours." Also, in the following section **In-situ surface-crosslinking**, it is mentioned that *each membrane in Cl⁻ form (10 × 12 cm) was ..."*

Furthermore, we also recognize that the reaction scheme in Figure 1a may cause some misunderstanding regarding the localization of the ISIP reaction. We emphasize that ISIP modification occurs exclusively on the membrane surface, while the internal membrane structure remains unchanged and is consistent with the bulk NML-Q membranes. The figure caption and related text have been revised to improve clarity, please see **Figure 1a**.

Q 6. At line 142, should it refer to Figure 3f instead of Figure 2f?

Response: Thank you for your careful observation. Upon review, we confirmed that the correct Figure reference at line 141 is Figure 3f. We have updated the manuscript accordingly. Please see line 119: *"the ED results shown in Figure 3b-f."*

Q 7. Some expressions appear somewhat lengthy. It would be beneficial to streamline them for better readability. For example, the sections introducing the three-phase membrane structure and describing phase separation could be simplified.

Response: Thank you for your constructive comment. We have rephrased the text in lines 120–133 to improve clarity and conciseness, as follows:

"The membrane's enhanced performance originates from its three-phase structure, comprising a hydrophobic region (PAB), an ionic phase (QA groups), and an auxiliary phase. In comparison, conventional ion exchange membranes (IEMs) typically exhibit a two-phase structure, consisting of a hydrophilic domain containing functional groups and a hydrophobic polymer backbone (Supplementary Fig. S4). The inclusion of the auxiliary phase fundamentally alters ion transport behavior. In standard IEMs, selectivity is achieved by allowing counterions to migrate through the hydrophilic phase, while co-ions are excluded due to electrostatic repulsion. Increasing the density of charged groups in conventional IEMs to enhance ion flux often results in excessive water uptake and swelling, which compromises membrane integrity and reduces selectivity—leading to a trade-off between flux and rejection. In contrast, the auxiliary phase introduced in this study, composed of –NH₂ groups, promotes microphase separation within the membrane, facilitating the transport of small ions while hindering the

passage of larger ones. This structural optimization enables high selectivity without sacrificing ion flux.”

Q 8. The authors are encouraged to clearly specify the conditions of electro dialysis in the figure captions to avoid any potential confusion.

Response: We appreciate this helpful suggestion and have updated the figure captions accordingly to clearly state ED conditions. Figure 3 has been updated to include the operation settings used during the electro dialysis tests, Please see line 294-297:

“The ED setup consists of two CEMs on either side of a central NML-Qx-NH₂ membrane with a membrane area of 7.07 cm². Ion transport is driven by an applied current density of 5 mA/cm². The electrolyte used is 0.3 mol L⁻¹ Na₂SO₄, with feed solutions of 0.1 mol L⁻¹ NaCl/Na₂SO₄, and 0.01 mol L⁻¹ KNO₃ (except for NO₃⁻ tests, where deionized water is used).” Additionally, the conditions for Figure 3f have been specified as: *“(f) Stability testing of NML-Qx-NH₂ membranes over 10 operational cycles performed in 0.1 mol L⁻¹ NaCl/Na₂SO₄ solution at a current density of 5 mA cm⁻² and a membrane area of 7.07 cm².”*

Q 9. How does the AFM image in Figure 2b reflect the microphase separation structure of the membrane? Given that the tested area corresponds to the membrane's surface layer (i.e., the ISIP layer), further clarification is recommended.

Response: We thank the reviewer for the valuable comments. The AFM images in **Figures 2a-b** reveal the microphase-separated structure at the membrane surface, originating from the intrinsic properties of the block copolymer and the quaternization process, not the ISIP process itself. Bright regions correspond to hydrophilic domains (e.g., quaternary ammonium groups), which absorb moisture/water and swell, while dark regions represent hydrophobic domains. As previously reported, the QPAB polymer shows a less defined morphology (low IEC), and as the IECs increase, this lead to a stronger phase contrast, highlighting well-separated domains. This structure plays a key role in enabling ion-selective transport [Ref 1].

After TMC modification, the AFM image retains these features with a slight roughness increase (**from 1.038 nm to 1.887 nm**), confirming mild and surface-localized TMC reaction without altering the original structure. SEM images show no thick layers or patterns typical of interfacial polymerization, indicating the microphase-separated morphology remains intact after TMC treatment.

Q 10. Figure 4i presents the results of a continuous electro dialysis process. Could the authors elaborate on how these results support the viewpoint mentioned at line 310?

Response: We thank the reviewer for the helpful comment. Upon carefully reviewing this section ourselves, we realized that the originally cited figure (**Figure 4i**) did not accurately support the statement previously located at line 310, now revised to lines 281–286. We apologize and have corrected the citation and now refer to Figure 3b, which more appropriately supports the intended explanation in the revised manuscript. “As previously reported by Li and Xu⁴², the hydration energy of SO_4^{2-} ions (ca. 1145 kJ mol⁻¹) is much higher than that of Cl^- ions (317 kJ mol⁻¹), indicating that SO_4^{2-} is more strongly hydrated and thus less likely to adhere to the membrane surface. Moreover, Sata et al.⁴³ also noted that membranes with hydrophobic surfaces have lower affinity for highly hydrated ions like SO_4^{2-} . These insights align well with the ion selectivity trends presented in Figure 3b.”

Q 11. Overall, the membranes described in the manuscript exhibit high ion fluxes. It is suggested that the authors calculate the current efficiency to demonstrate that such high ion fluxes fall within a reasonable range.

Response: We thank the reviewer for this suggestion. We have calculated the current efficiency (%) for each membrane based on the measured Cl^- ion flux and the number of moles transported. The current efficiency (%) was determined using the concentrations and charges of the transported ions, the applied current and duration, the solution volume, and Faraday's constant, as shown in Eq (6) in the revised manuscript [Ref 2]:

$$\text{Current Efficiency } (\eta \%) = \left(\frac{z_i F V \Delta C_i}{I t} \right) \times 100 \quad (6)$$

where ΔC_i is the change in ion concentration (mol L⁻¹), z_i is the ion charge, F is Faraday's constant (C mol⁻¹), V is the volume of solution (L), I is the current (A), and t is time (s).

Please see supplementary Table S4: Cl^- flux, moles transported, and current efficiency for different membrane samples (NML, NML-Q₁, NML-Q₂, NML-Q₃, NML-Q₁-NH₂, NML-Q₂-NH₂, and NML-Q₃-NH₂) during ED.

Sample	Cl ⁻ ion flux (mol m ⁻² h ⁻¹)	Moles Transported	Current Efficiency (%)
NML	0.9584	0.000678	50.47
NML-Q ₁	1.05745	0.0007475	55.67
NML-Q ₂	1.34739	0.0009529	70.94
NML-Q ₃	1.80489	0.001275	94.89
NML-Q ₁ -NH ₂	2.09373	0.001480	110.20
NML-Q ₂ -NH ₂	1.98314	0.001402	104.40
NML-Q ₃ -NH ₂	2.00231	0.001415	105.35

It is important to note that current efficiency (%) values exceeding 100% may initially appear unusual; however, such values have been documented in previous studies and can arise under specific operating conditions. For example, Luo et al. used a specialized ED setup incorporating a cation exchange membrane (CEM) to concentrate formic acid. When the feed concentration was high (approximately 10 wt% formic acid), a CE greater than 100% was observed [Ref 3]. In another study, Lysova and co-workers reported a current efficiency of 197% for the Neosepta commercial membrane when separating monovalent ions like Cl⁻ and SO₄²⁻ [Ref 2]. Similarly, Nidal et al. developed a hybrid cation-exchange membrane by incorporating polybenzimidazole (PBI) into a Nafion-117 matrix via in situ polymerization. They also reported a high current efficiency exceeding 100%, along with improved monovalent ion selectivity [Ref 4]. These examples show that CEs values above 100% are not unprecedented and may also occur in our system. Such elevated CE values can result from factors such as water transport across the membrane, co-ion leakage, or additional ion movement to maintain charge balance, which may collectively lead to an apparent ion transfer exceeding that predicted by the applied current.

Q 12. The authors are kindly advised to carefully review the description of ion flux values from lines 450 to 457. Did the authors calculate the ion flux without considering the time factor? Ideally, the ion flux should remain relatively stable over time rather than increasing significantly.

Response: We thank the reviewer for the opportunity to clarify this point. We would like to clarify that the values reported in Figure 4i represent the *transported amount* of Cl⁻ over time,

not instantaneous flux rates. We confirm that the flux J_i ($\text{mol m}^{-2} \text{ h}^{-1}$) of the anions was calculated using the following Eq (4) in the revised manuscript:

$$J_i = \frac{(C_t - C_i) \cdot V}{A_m \cdot t} \quad (4)$$

V is the volume (0.1 L), C_t and C_i represent the final and initial molar concentration (mol L^{-1}) in the concentrated compartment, t is the time (hour), A_m is the effective area (m^2) of membrane.

Please see line 435-442.

“In Figure 4i, the 1-hour value (1.90 mol m^{-2}) represents the amount of Cl^- transported during the first hour, while the 2-hour (3.87 mol m^{-2}) and 3-hour (6.44 mol m^{-2}) values reflect the cumulative transport over two and three hours, respectively. Since these are cumulative values, an increasing trend over time is expected. Importantly, the calculated flux rates ($1.90 - 6.44 \text{ mol m}^{-2} \text{ h}^{-1}$) remain relatively steady, indicating stable transport behaviour. Additionally, the SO_4^{2-} transport remains minimal throughout the test (0.075 to 0.28 mol m^{-2}), and the $\text{Cl}^-/\text{SO}_4^{2-}$ selectivity remains high (25.31 to 22.77), demonstrating consistent separation performance during the entire operation.”

Q 13. In the experimental section, DPAB is used to represent the deprotonated polymer, whereas PAB is utilized in the main text. A proofreading check is recommended to ensure consistency.

Response: Thank you for your helpful comment. To improve clarity, we now use a single abbreviation (PAB) for the deprotonated polymer throughout the manuscript. This change has been made in the revised version (see line 542).

Reviewer 2- Evaluation

General comment: The manuscript titled “Performance Evaluation of Monovalent Ion-Selective Membranes Fabricated via In-Situ Interfacial Polymerization (ISIP) for Electrodialysis Applications,” presents a novel and promising approach to monovalent anion-selective membranes. The work demonstrates impressive selectivity, supported by DFT/MD simulations. However, significant issues related to data reproducibility, methodological gaps, and structural organization currently (redundancy, missing schemes, figure mislabeling, and unit notation) limit the robustness of the study. There are also concerns regarding clarity of

experimental reporting with the lack of discussion. Major revisions are required before the manuscript can be reconsidered for publication to enhance clarity, reproducibility, and impact so that it meets the journal's standards.

Response: Thank you very much for your helpful feedback and for recognizing the novelty and usefulness of our ISIP-based membrane method. We have carefully considered all your comments and made several improvements to the manuscript accordingly. Revisions were undertaken to improve data reproducibility, clarify methodologies, and enhance the overall structure and organization of the manuscript. All figures have been redrawn; labels, layouts, and units have been revised, and missing elements have been added as suggested. The schemes have also been updated to more clearly illustrate the material preparation process. Additionally, we have revised the experimental section for greater clarity and expanded the discussion to better explain the significance of our findings. Please see our point-by-point responses below.

Q 1: In the introduction section, the statement "To address these limitations, research has focused on developing advanced anion-selective membranes." should be corrected to "To address these limitations, research has focused on developing advanced anion-selective membranes by incorporating materials such as zeolites [11], metal-organic frameworks (MOFs) ...", since these materials are components, not membranes themselves.

Response: Thank you for your thoughtful comment. You are absolutely right that materials like zeolites, MOFs, and COFs are generally used as components within membranes, rather than as membranes themselves. We agree that this distinction is important, and we have revised the sentence in the introduction accordingly to better reflect this point (please see revised lines 50-55).

"To address these limitations, research has focused on developing advanced anion-selective membranes by incorporating functional materials such as zeolites [11], metal-organic frameworks (MOFs) [12], and covalent organic frameworks (COFs)[13]. While these materials show promise due to their ion selectivity and transport efficiency, challenges such as high synthesis costs, scalability, and instability in aqueous environments hinder their widespread use."

Q 2. The introduction mentions: "Despite these efforts, none of these methods have successfully addressed the critical challenges of achieving low cost, high selectivity, and scalability

simultaneously. These limitations highlight the urgent need for a more innovative and integrated approach to membrane design.” While the study presents a novel approach, the claims regarding cost-effectiveness and scalability remain speculative and overstated. The synthesis process, including custom polymer synthesis, quaternization, and surface modification, appears to be a lab-intensive, multi-step process. Since the scalability/cost claims are unsupported, the manuscript would benefit from an economic analysis, or the sentence should be reworded to reflect the potential challenges in scaling up this approach.

Response: Thank you for your suggestion. We understand that the original sentence may have given the impression that our method is already proven to be low-cost and scalable. Based on the reviewer comment we have revised the sentence. The revised sentence in the manuscript can be found at lines 73-78

“Despite these efforts, none of these methods have successfully addressed the key challenges of achieving high selectivity, operational stability, and practical scalability together. While some techniques offer improvements in specific areas, a complete solution that meets both performance and manufacturing needs is still lacking. This highlights the need for a more integrated approach to membrane design that not only focuses on lab performance but also considers future application, cost, and scale-up challenges.”

Q 3. In the materials section, the abbreviation for cation exchange membranes is given as “CM,” but under Figure 3 it is referred to as “CEM.” Please ensure consistent use of all abbreviations throughout the manuscript.

Response: Thank you for pointing this out. We have corrected the abbreviation 'CM' to 'CEM'; please refer to the Materials section at line 528. We have also reviewed the entire manuscript for similar typographical errors.

Q 4. The molar ratio of 1:1.34 between biphenyl and 4-acetylpyridine appears intentional. Please clarify the rationale behind this specific ratio. Was it experimentally optimized to control chain-end functionality or another property?

Response. Thank you for your insightful comment. Yes, the use of a 1:1.34 molar ratio between biphenyl and 4-acetylpyridine was intentional and follows the principles of non-stoichiometric (A_2+B_2) step-growth polymerization, which is commonly used to control chain-end functionality and improve molecular weight in superacid-catalyzed systems. Specifically, we

employed a slight excess of the 4-acetylpyridine (B) to favor linear polymer growth. Previous studies, including our own [Ref 1], have reported similar molar ratios to achieve these goals. Similarly, Cetina-Mancilla and Zolotukhin demonstrated that non-stoichiometric feed ratios are essential for obtaining well-defined aromatic polymers with high molecular weight and controlled pyridine content [Ref 5]. Also, Stonawski et al. reported that even a ~10% excess of the ketone monomer significantly improved polymer chain growth [Ref 6]. In conclusion, the slight excess of 4-acetylpyridine is a deliberate design choice that enables high-performance polymer formation with well-defined chain architecture, enhanced solubility, and suppressed gelation key for successful superacid-mediated polymerization.

Q 5. In the Experimental section, Scheme 1a and Scheme 1b are referenced, but the schemes themselves are not provided in the manuscript or supplementary file. Please include these schemes for completeness.

Response: We appreciate the reviewer's careful observation. We have corrected the text in the Experimental section. The processes originally referred to as Scheme 1a and Scheme 1b are now correctly cited as Figure 1a, which illustrates both the preparation of Poly(alkyl-biphenyl pyridine) (PAB) and the quaternization of the PAB polymer. This figure was already included in the main manuscript, and we have revised the references accordingly to maintain consistency and clarity.

Q 6. The sentence "During this period, the solution gradually developed a purplish-red color, as illustrated in Scheme 1a." refers to a scheme that could not be found. Kindly correct the reference or include the relevant illustration.

Response: Thank you for pointing this out. We apologize for the oversight. The sentence originally referring to "Scheme 1a" has been corrected to "Figure 1a" in the revised manuscript. Additionally, this correction was already addressed in our response to Comment 5. We have carefully reviewed the manuscript to ensure consistency and have corrected similar referencing errors where applicable.

Q 7. The manuscript references "Various amounts of amine, as shown in Table 1..." but the table is in the supplementary material. The reference should be revised to "Table S1" for consistency with the rest of the manuscript.

Response: Thank you for your observation. We have corrected and the sentence now states: “*Various amounts of amine, as shown in Table S1...*” (see line 545). This clarifies that the table is part of the Supporting Information.

Q 8. The manuscript uses both " $\text{mol}\cdot\text{L}^{-1}$ " and " mol L^{-1} ", or " $\text{mmol}\cdot\text{g}^{-1}$ " " mmol/g " and so on. For consistency, uniform unit notation should be used throughout the manuscript.

Response: Thank you for your comment. We have now used the same style for all units in the paper. For example, we changed " $\text{mol}\cdot\text{L}^{-1}$ " to " mol L^{-1} " and " $\text{mmol}\cdot\text{g}^{-1}$ " to " mmol g^{-1} ".

Q 9. The manuscript states that the polymer was stirred in 1 M NaOH for 24 hours for complete deprotonation. Was the pH of the supernatant monitored after the treatment, or was fresh NaOH added during deprotonation process? Please clarify these details to ensure complete deprotonation.

Response: Thank you for your valuable comment. The deprotonation procedure we used is consistent with previously reported protocols for pyridine-containing polymers, including our own prior work [Ref 1]. We did not monitor the pH of the supernatant or add fresh NaOH during the 24-hour treatment, as the amount of protonated pyridine groups in the polymer is very low compared to the large excess of 1 mol L^{-1} NaOH used. Given this stoichiometric excess and the strong basicity of NaOH, it is standard practice in the field not to monitor pH or replenish the base during the reaction.

Q 10. The procedure for membrane casting after quaternization is not described. Please provide information on the casting method, including casting thickness and drying conditions, to complete the experimental protocol.

Response: Thank you for your constructive comment. The requested information has been added to the manuscript as follows (line 546-554):

“The membrane was prepared using the solution casting. After the quaternization reaction, the polymer solution was filtered through a $0.5 \mu\text{m}$ syringe filter and cast onto a clean glass plate, with the edges sealed using $\sim 0.1 \text{ mm}$ thick transparent insulation tape to prevent outflow. The cast film was dried at $60 \text{ }^\circ\text{C}$ for 6-7 hours to fully evaporate the solvent. The dried membrane was then gently detached by soaking in water, thoroughly washed, and converted to the Cl⁻ form by immersing it in a 0.5 mol L^{-1} NaCl solution for 24 hours. For comparison, a reference

membrane (NML) was prepared using the same method without amine addition, meaning no quaternization occurred”.

Q 11. In addition to measuring water uptake, swelling ratio or dimensional changes (such as length or thickness) should be included to provide a clearer understanding of the membrane's structural stability.

Response: Thank you for your constructive feedback. In response, the swelling ratio has been added to manuscript. Additionally, the procedure used to measure the swelling ratio has been detailed as follows:

“The swelling ratio (SR) was calculated using Eq 3 as the area difference between wet (S_{wet} , cm^2) and dry (S_{dry} , cm^2) membrane samples.

$$SR(\%) = \frac{S_{wet} - S_{dry}}{S_{dry}} \times 100 \% \quad (3)$$

Q 12. The ED setup is incorrectly referenced in Figure 1a, which shows synthesis steps of the polymer. Please correct the figure reference.

Response: Thank you for your comment. We fixed the error. The correct figure for the ED setup is Figure 3a. Please see the change on line 615.

Q 13. In Figure 1a, DMC is mentioned without being defined. Please expand the abbreviation (e.g., dichloromethane or CH_2Cl_2) for clarity.

Response: Thank you for your comment. We have now written “DCM” as the short form of dichloromethane in the text. We also removed the chemical formula CH_2Cl_2 from the materials section to match what is shown in the synthesis Figure 1a. Please see the change on lines 525–526.

Q 14. Please provide units for the variables C_t , C_i (initial and final molar concentrations), and time (t) in the relevant equations to ensure clarity and reproducibility.

Response: Thank you for your valuable suggestion. As per your suggestion, we have updated Eq (4) in the revised manuscript (please see line 663) to clearly define the units for all variables. The changes are as follows:

The flux J_i ($\text{mol m}^{-2} \text{h}^{-1}$) of the anions was calculated using the following Eq (4):

$$J_i = \frac{(C_t - C_i) \cdot V}{A_m \cdot t} \quad (4)$$

V is the volume (0.1 L), C_t and C_i represent the final and initial molar concentration (mol L^{-1}) in the concentrated compartment, t is the time (hour), A_m is the effective area (m^2) of membrane.

Q 15. Equation (4) states that “ C_t and C_i represent the initial and final molar concentrations,” but by convention, C_t typically denotes the concentration at time t (i.e., final concentration), and C_i refers to the initial concentration. Please revise this for clarity.

Response: Thank you for your comment. Yes, it was a typing mistake. We have now revised the notation to align with standard conventions, please see line 628 “ C_t and C_i represent the final and initial molar concentration (mol L^{-1}) in the concentrated compartment.”

Q 16. Radar plots in Figure S3 lack error bars and numerical precision. Please specify how many times each experiment was repeated. A comparative table with actual values and standard deviations would significantly improve the clarity of data.

Response: Thank you for your constructive comment. Each experiment was conducted at least three times. We have now added a table along with the radar plots in the Supporting Information.

Table 2. Water uptake (WU), ion exchange capacity (IEC), transport number, membrane resistance (R_m), limiting current density (LCD), and thickness (μm) for different membranes

Membrane	WU (%)	IEC (mmol g^{-1})	Transport Number	R_m ($\Omega \text{ cm}^2$)	LCD (mA cm^{-2})	Thickness (μm)
NML-Q ₁	4.21 ± 0.21	0.34 ± 0.02	0.76 ± 0.04	–	–	24 ± 1.20
NML-Q ₂	16.96 ± 0.85	0.62 ± 0.03	0.83 ± 0.04	3.82 ± 0.19	112 ± 6	26 ± 1.30
NML-Q ₃	30.04 ± 1.50	1.00 ± 0.05	0.88 ± 0.04	2.31 ± 0.12	114 ± 6	26 ± 1.34
NML-Q ₁ -NH ₂	5.68 ± 0.28	0.29 ± 0.01	0.71 ± 0.04	–	–	29 ± 1.45
NML-Q ₂ -NH ₂	18.17 ± 0.91	0.65 ± 0.03	0.76 ± 0.04	7.4 ± 0.37	91 ± 5	23 ± 1.45

Membrane	WU (%)	IEC (mmol g ⁻¹)	Transport Number	R _m (Ω cm ²)	LCD (mA cm ⁻²)	Thickness (μm)
NML-Q3-NH ₂	32.98 ± 1.65	0.88 ± 0.04	0.81 ± 0.04	4.06 ± 0.20	101 ± 5	26 ± 1.35

* *We added a new table, therefore the numbers of all tables have been changed. We marked the new table numbers in red in the Supporting Information to make them easy to find.*

Q 17. The manuscript states that “after surface modifications, the IEC of the membranes decreased from NML-Q1-NH₂ to NML-Q₃-NH₂,” but Figure S3 shows NML-Q₃-NH₂ with a higher IEC. Please clarify this inconsistency or restate the sentence.

Response: Thank you for this careful observation. After double-checking the data and Figure S3, we would like to clarify that the IEC of NML-Q₃-NH₂ is indeed lower than that of NML-Q₃ (prior to modification). We suspect there may have been a misunderstanding in interpreting the Figure S3. For clarity, we have now restated the sentence in the revised manuscript (lines 104–105) as follows: 103-104 “*After surface modifications, the IEC values showed an overall decreasing trend, except for NML-Q₃-NH₂.*”

Q 18. The manuscript mentions that “COOH groups serve as active functional sites, reducing IEC through electrostatic interactions with the oppositely charged QA groups.” However, while the IEC of NML-Q1-NH₂ is lower than that of NML-Q1, the IEC of NML-Q₂-NH₂ and NML-Q₃-NH₂ is higher than their respective non-modified counterparts. Please address this discrepancy and make a clear explanation for the relationship between surface modification and ion exchange capacity.

Response: Thank you for the comment. The experimental data show that only NML-Q₂-NH₂ has a slightly higher IEC than its non-modified counterpart. To clarify this, we have revised the manuscript text. As mentioned in our response to **Comment 17**, please see lines 103-104 “*After surface modifications, the IEC values showed an overall decreasing trend, except for NML-Q₃-NH₂.*”

Q 19. NML-Q₃-NH₂ exhibits higher resistance (R_m) than NML-Q₃ despite having higher IEC and water uptake. Is this due to surface modification? Cross-section SEM image should be provided to show the thickness of the ISIP-formed layer and clarify this point. Additionally, thicknesses of each membrane are recommended to be tabulated before and after surface modification.

Response: Thank you for your helpful comment. We would like to clarify that the IEC of NML-Q₃-NH₂ is actually lower than that of NML-Q₃ (0.88 vs. 1.00 mmol g⁻¹), not higher. Although NML-Q₃-NH₂ exhibits slightly higher water uptake (32.98% vs. 30.04%), this likely indicates that additional water is retained through interactions with –COOH groups introduced during surface modification. The increased resistance (from 2.31 to 4.06 Ω cm²) after modification is likely due to surface –COO⁻ groups and mild crosslinking.

We would like to clarify that our method differs from traditional interfacial polymerization (IP), where two monomers react to form a distinct polymer layer on the membrane surface. In our approach, only TMC is used, which reacts selectively with –NH₂ groups already present on the membrane surface. As a result, no separate surface layer is expected to form. This is supported by the SEM images (Figure S7), which do not show any distinct or additional layer after the reaction (also see Figure 3). Regarding the small differences in membrane thickness reported in Table S2, we note that these variations are not due to the ISIP process. All membranes were prepared on a glass substrate, and minor thickness variations can occur due to manual dry casting or slight unevenness in the hot plate. These small differences are common and not related to the surface modification itself.

Figure 3. Cross-sectional SEM images of NML-Q₁, NML-Q₂, and NML-Q₃ samples before (top row) and after surface modification with TMC, shown as NML-Q₁-NH₂, NML-Q₂-NH₂, and NML-Q₃-NH₂ (bottom row).

Q 20. The section “Optimizing ion transport channels” discussing ion transport mechanisms in traditional IEMs is redundant in this manuscript, as the focus is on anion permselectivity. Consider moving this background to Supplementary Information (e.g., under Supplementary Figure S4) to streamline the main text and remove repetitive statements.

Response: Thank you for the helpful suggestion. We have deleted the section on ion transport mechanisms to avoid repetition. The following revision has been made to the results section:

“NML-Q membranes for ion separation

The NML membranes were designed by functionalizing a hydrophobic PAB-based backbone with varying amounts of QA groups to enhance IEC and membrane hydrophilicity. The pristine NML membrane, lacking QA groups, displayed negligible WU and was therefore characterized only after functionalization.”

Q 21. Figure 4i does not depict hydration energies as described in the manuscript; it shows flux and selectivity. Please clarify.

Response: Thank you for pointing this out. Figure 3g has been corrected to match the description in the text. It now accurately reflects hydration energies.

Q 22. High water uptake with high number of quaternary ammonium groups typically favor multivalent over monovalent ions, as hydrophilicity of the membranes increase. How do the authors explain the higher Cl^- selectivity over SO_4^{2-} in NML-Q2 and NML-Q3 membranes? Water contact angle measurements are strongly recommended to justify surface hydrophilicity/hydrophobicity as being stated in the manuscript and clarify the interaction of ions with varying hydration energies.

Response: Thank you very much for your thoughtful comment. In our case, despite the increased water uptake and IEC, Cl^- selectivity remains higher due to its smaller size, lower charge, and weaker hydration shell compared to SO_4^{2-} . These properties allow Cl^- to move more easily through the membrane’s ionic channels, while SO_4^{2-} , being bulkier and more strongly hydrated, is more restricted. This behavior is supported by our data (Table S5, Fig. 4g) and aligns with previous literature [Ref 7, 8].

The following statements were added to the manuscript:

(Line 577-579): “Static water contact angles were measured at room temperature using an Ossila Goniometer with a 0.47 mm blunt needle. Values are reported as mean \pm standard deviation from multiple measurements”

(Line 278-281): “Another factor influencing ion flux is membrane hydrophilicity. As shown in Table S3, increased swelling ratio and decreased contact angle from NML-Q₁ to NML-Q₃ reflect improved water uptake and surface wettability, which support higher Cl⁻ transport.”

Q 23. The permselectivity results are very impressive and surpass many reported in the literature. It is noteworthy that the high concentration gradient used (0.1 mol L⁻¹ NaCl + Na₂SO₄ vs 0.01 mol L⁻¹ KNO₃) likely enhances monovalent ion transport, making it difficult to distinguish the true contribution of membrane properties or it is together with process conditions. To ensure a fair and accurate comparison, the authors should report the testing conditions (current density and solution concentrations) for each study in Table S3. This clarification is necessary to properly assess the exceptional performance of the NML-Q₂-NH₂ membrane.

Response: Thank you very much for your valuable comment. You are absolutely correct that the concentration gradient we used can influence ion transport through the membrane. We selected a mixed salt solution to better reflect real-world conditions, where membranes typically encounter complex ionic mixtures. To make sure the comparison is fair and transparent, we have updated the Table to include the current density and solution concentrations for each referenced study. We hope this addition improves the clarity and comparability of results.

Q 24. The energy consumption calculations per unit of ions separated should be included along with the permselectivity values in Table S3, as this is a critical metric for potential scalability concerns.

Response: We appreciate the reviewer’s insightful comment. We have now included the calculated energy consumption per mole of ion (kWh mol⁻¹) for the mixed ion solution (0.1 M NaCl/Na₂SO₄/NaNO₃/NaF/NaBr) in the revised Supporting Information (please see Table S6b).

Table 5b. Energy consumption (kWh mol⁻¹) of ion for a mix solution (0.1 M NaCl/Na₂SO₄/NaNO₃/NaF/NaBr) using NML-Q_x-NH₂.

Ion	NML-Q1-NH ₂	NML-Q2-NH ₂	NML-Q3-NH ₂
Cl ⁻	1.36	1.08	0.57
SO ₄ ²⁻	12.72	63.61	15.93
Br ⁻	1.00	1.08	1.08
NO ₃ ⁻	0.76	0.74	0.94
F ⁻	4.76	5.31	4.90

Q 25. It is worth adding the performance results obtained for Cl/SO₄ (>100) in the following study “Journal of Membrane Science 716 (2025) 123518” in Table S3.

Response: Thank you for the helpful suggestion. We reviewed the article and found it has valuable data on high selectivity. We added this study as *Ref 18* in the Supporting Information and included its performance results.

Q 26. The manuscript states “For example, SO₄²⁻, with a larger Stokes radius (2.31 Å), faces more resistance compared to Cl⁻ (1.21 Å).” Please provide reference for this information.

Response: We appreciate this constructive comment. We have now added the appropriate reference to support this statement. The reported Stokes radii of common anions are well-documented in the literature. For example, Abarkan et al. reported a Stokes radius of 2.30 Å for SO₄²⁻ and 1.21 Å for Cl⁻ [Ref 9]. Their work clearly explains that ions with larger Stokes radii, such as SO₄²⁻, experience higher resistance during transport due to greater hydrodynamic drag. This information has been cited (please see *Ref 42* in the revised manuscript).

Q 27. Please provide a separate table listing Stokes radii and hydration energies for all ions studied with corresponding references for readability and easier comparison of the permselectivity values.

Response: Thank you for the helpful comment. A new table (Table S7) has been added to the Supporting Information. It lists the Stokes radii and hydration energies of all studied ions, along with their references.

Supplementary Table S7

Table 5 represents the Ionic diameter, hydrated diameter, and hydration energy of anions tested in work.

Ion	Ionic Diameter (nm)	Hydrated Diameter (nm)	Hydration Energy (kJ mol ⁻¹)	Ref.
F ⁻	0.119	0.35	580	10
Cl ⁻	0.181	0.33	355	11
NO ₃ ⁻	0.189	0.34	328	10
Br ⁻	0.196	0.33	358	10
SO ₄ ²⁻	0.29	0.379	1145	10

Reviewer #3 (Remarks to the Author):

General comment: In this work, MD simulations were performed to examine the transport and interactions of ions within the polymer system. The authors need further clarification on the following questions.

Response: We are grateful for your input and have replied to each comment individually as outlined below.

Q 1. In the MD simulation methods, what is the meaning of ‘a small excess of the resp.’

Response: Thank you for your comment. This is a typographical mistake. We corrected the mistake.

“The bulk water pore system was simulated with both Cl⁻ and SO₄²⁻ ions present. A small excess of the respective Na⁺ salt corresponding to 0.08 mol L⁻¹ was used in all systems to improve sampling.”

Q 2. How is the electric field set during the simulation?

Response: Thank you for your comments. No external electric field was applied during the simulations presented in this manuscript, in order to reflect the intrinsic ion conduction behavior of the membrane. This represents a limiting condition, particularly relevant when the current density approaches zero. This approach is consistent with numerous published studies that have conducted similar simulations without the application of an electric field .

We would like to highlight that applying an electric field in simulations using periodic boundary conditions (PBC) is known to introduce artifacts in the diffusive motion of ions. This occurs

due to field-induced drift and unphysical charge compensation, which can distort both the applied field and ion–ion interactions. In GROMACS, the uniform neutralizing background required for long-range electrostatics (PME), along with finite-size and dielectric artifacts inherent to repeating PBC cells, leads to a situation where mean-square-displacement analysis no longer reflects true molecular diffusion. As a result, diffusion coefficients extracted under these conditions are likely to be systematically biased.

To provide the reviewer with a clearer understanding of this limitation, we also ran simulations under medium and high electric field strengths. As expected, the diffusion coefficients obtained in these cases were systematically biased. For this reason, we chose not to include these results in the manuscript.

Figure 4. MD simulations were used to study the effect of an external electric field (along Z) on ion and water transport in a bulk pore system. (a) Normalized diffusion coefficients (D) for Na^+ , Cl^- , SO_4^{2-} , and water under field strengths of 0, 1×10^{-7} , and 1×10^{-4} V nm^{-1} . (b) The simulation box showing the polymer matrix and distribution of water and ions. Simulations were run for 200 ns in the NVT ensemble using GROMACS. Diffusion was calculated with *gmx msd*, using polymer-centered trajectories to avoid net drift.

Q 3. Is the simulation system maintaining neutral? How is the neutrality maintained?

Response: Thank you for your interest in the simulation details. Yes, all simulation systems were charge-neutral, with extra Na^+ ions used to balance the slight excess of anions.

Q 4. The exact procedure for calculating the free energy profiles needs to be given in detail. How are the Boltzmann-weighted density profiles obtained?

Response: The free energy profiles were derived from the position-dependent ion density profiles across the polymer-water interface using the Boltzmann inversion (Eq (S1)):

$$\frac{\Delta G}{kT} = -\ln\left(\frac{[C]}{[C_0]}\right) \quad (S1)$$

In Eqn (S1), [C] is the local ion concentration at position zzz, and [C₀] is the reference ion concentration in the bulk pore region, outside the polymer. This approach assumes that the system is at equilibrium and that the ion distributions follow Boltzmann statistics. The ion density profiles, were obtained by binning the simulation trajectory data along the direction normal to the polymer interface, and averaging the number of ions in each bin over the simulation time. The resulting free energy profiles provide insight into the thermodynamic favorability of ion partitioning between the aqueous and polymer regions. The results show that larger, more hydrated ions such as SO₄²⁻ exhibit higher free energy barriers within the polymer phase compared to smaller ions like Cl⁻, supporting the observation that such ions are more strongly retained and less mobile within the polymer matrix.

The equation has been added to the Supplementary Information as Eq (S1), along with a detailed explanation of the calculation procedure:

“Free energy profiles were obtained from the density profiles of the ions across the polymer-water phase (Figure 3g), using the Boltzmann inversion (Eq (S1)):

$$\frac{\Delta G}{kT} = -\ln\left(\frac{[C]}{[C_0]}\right) \quad (S1)$$

Where [C] is the position-dependent ion concentration and [C₀] the corresponding ion reference concentration in the bulk pore outside the polymer. The free energy results support the idea that larger and more hydrated ions like SO₄²⁻ are more likely to interact and be retained in the polymer matrix, thus impacting their transport more significantly than smaller ions like Cl⁻.”

Q 5. ‘due to its larger hydrated radius and stronger interactions with the membrane surface (hydrophilicity)’, please support the above conclusion with data from molecular simulations.

Response: We appreciate this constructive comment. We followed the reviewer comments and modified the sentence as follow (line 324-328):

“This aligns well with the observation that SO_4^{2-} ion mobility through the membrane was lower compared to Cl^- , due to its larger hydrated radius, which from a radial distribution analysis was found to be 0.358 nm, compared to 0.315 nm for Cl^- . This is also well aligned with the stronger interactions with the membrane surface, demonstrated by the lower free energy shown in Figure 3g.”

The figure below presents the results of a radial distribution function (RDF) analysis for hydrated SO_4^{2-} and Cl^- ions. The data indicate that SO_4^{2-} exhibits a larger hydrated radius compared to Cl^- , as well as a higher coordination number within its first solvation shell, due to its greater charge density and higher overall hydration energy.

Figure 5. RDF and CN analysis of anionic solutes with water oxygen atoms. (a) RDF data for S (in SO_4^{2-}) and Cl^- demonstrating the larger hydrodynamics radii of SO_4^{2-} compared to Cl^- (b) Accumulated coordination number (CN) for SO_4^{2-} and Cl^- . Dashed lines indicate first coordination shell limits, corresponding to CN being approx. 14 and 8 for SO_4^{2-} and Cl^- , respectively.

References

1. Yang H, Afsar NU, Chen Q, Ge X, Li X, Ge L, Xu T. Poly(alkyl-biphenyl pyridinium) anion exchange membranes with a hydrophobic side chain for mono-/divalent anion separation. *Industrial Chemistry & Materials* **1**, 129-139 (2023).
2. Lysova AA, Manin AD, Golubenko DV, Ponomarev II, Altynov VA, Hilal N, Yaroslavtsev AB. Ultra-high nitrate-selective metal-polymer membranes based on cardo polybenzimidazole for electro dialysis. *Journal of Membrane Science* **716**, 123518 (2025).
3. Luo GS, Pan S, Liu JG. Use of the electro dialysis process to concentrate a formic acid solution. *Desalination* **150**, 227-234 (2002).
4. Golubenko D, Petukhov D, Al-Juboori RA, Johnson D, Hilal N. Polybenzimidazole-Modified Cation-Exchange Membrane with High Monovalent Ion Selectivity for Electro dialysis Separation of Alkaline/Alkaline Earth Metals. *ACS Applied Polymer Materials* **6**, 11762-11775 (2024).
5. Cetina-Mancilla E, Olvera LI, Balmaseda J, Forster M, Ruiz-Treviño FA, Cárdenas J, Vivaldo-Lima E, Zolotukhin MG. Well-defined, linear, wholly aromatic polymers with controlled content and position of pyridine moieties in macromolecules from one-pot, room temperature, metal-free step-polymerizations. *Polymer Chemistry* **11**, 6194-6205 (2020).
6. Guzmán-Gutiérrez MT, Nieto DR, Fomine S, Morales SL, Zolotukhin MG, Hernandez MCG, Kricheldorf H, Wilks ES. Dramatic Enhancement of Superacid-Catalyzed Polyhydroxyalkylation Reactions. *Macromolecules* **44**, 194-202 (2011).
7. Li J, Xu Z, Liao J, Ang EH, Chen X, Mu J, Shen J. Revolutionary MOF-enhanced anion exchange membrane for precise monovalent anion separation through structural optimization and doping. *Desalination* **576**, 117352 (2024).
8. Ruan H, Pan N, Wang C, Yu L, Liao J, Shen J. Functional UiO-66 Series Membranes with High Perm Selectivity of Monovalent and Bivalent Anions for Electro dialysis Applications. *Industrial & Engineering Chemistry Research* **60**, 4086-4096 (2021).
9. Abarkan A, Grimi N, Métayer H, Sqalli Houssaini T, Legallais C. Electro dialysis Can Lower the Environmental Impact of Hemodialysis. *Membranes* **12**, 45 (2022).
10. Tansel B. Significance of thermodynamic and physical characteristics on permeation of ions during membrane separation: Hydrated radius, hydration free energy and viscous effects. *Separation and Purification Technology* **86**, 119-126 (2012).
11. David F, Vokhmin V, Ionova G. Water characteristics depend on the ionic environment. Thermodynamics and modelisation of the aquo ions. *J Mol Liq* **90**, 45-62 (2001).

Evaluation

The manuscript titled “Performance Evaluation of Monovalent Ion-Selective Membranes Fabricated via In-Situ Interfacial Polymerization (ISIP) for Electrodialysis Applications,” presents a novel and promising approach to monovalent anion-selective membranes. The work demonstrates impressive selectivity, supported by DFT/MD simulations. However, significant issues related to data reproducibility, methodological gaps, and structural organization currently (redundancy, missing schemes, figure mislabeling, and unit notation) limit the robustness of the study. There are also concerns regarding clarity of experimental reporting with the lack of discussion. Major revisions are required before the manuscript can be reconsidered for publication to enhance clarity, reproducibility, and impact so that it meets the journal’s standards.

Comments

Comment 1: In the introduction section, the statement "To address these limitations, research has focused on developing advanced anion-selective membranes..." should be corrected to “To address these limitations, research has focused on developing advanced anion-selective membranes **by incorporating materials** such as zeolites [11], metal-organic frameworks (MOFs) ...”, since these materials are components, not membranes themselves.

Comment 2. The introduction mentions: “Despite these efforts, none of these methods have successfully addressed the critical challenges of achieving low cost, high selectivity, and scalability simultaneously. These limitations highlight the urgent need for a more innovative and integrated approach to membrane design.” While the study presents a novel approach, the claims regarding cost-effectiveness and scalability remain speculative and overstated. The synthesis process, including custom polymer synthesis, quaternization, and surface modification, appears to be a lab-intensive, multi-step process. Since the scalability/cost claims are unsupported, the manuscript would benefit from an economic analysis, or the sentence should be reworded to reflect the potential challenges in scaling up this approach.

Comment 3. In the materials section, the abbreviation for cation exchange membranes is given as “CM,” but under Figure 3 it is referred to as “CEM.” Please ensure consistent use of all abbreviations throughout the manuscript.

Comment 4. The molar ratio of 1:1.34 between biphenyl and 4-acetylpyridine appears intentional. Please clarify the rationale behind this specific ratio. Was it experimentally optimized to control chain-end functionality or another property?

Comment 5. In the Experimental section, Scheme 1a and Scheme 1b are referenced, but the schemes themselves are not provided in the manuscript or supplementary file. Please include these schemes for completeness.

Comment 6. The sentence “During this period, the solution gradually developed a purplish-red color, as illustrated in Scheme 1a.” refers to a scheme that could not be found. Kindly correct the reference or include the relevant illustration.

Comment 7. The manuscript references “Various amounts of amine, as shown in Table 1...” but the table is in the supplementary material. The reference should be revised to “Table S1” for consistency with the rest of the manuscript.

Comment 8. The manuscript uses both " $\text{mol}\cdot\text{L}^{-1}$ " and " mol L^{-1} ", or " $\text{mmol}\cdot\text{g}^{-1}$ " " mmol/g " and so on. For consistency, uniform unit notation should be used throughout the manuscript.

Comment 9. The manuscript states that the polymer was stirred in 1 M NaOH for 24 hours for complete deprotonation. Was the pH of the supernatant monitored after the treatment, or was fresh NaOH added during deprotonation process? Please clarify these details to ensure complete deprotonation.

Comment 10. The procedure for membrane casting after quaternization is not described. Please provide information on the casting method, including casting thickness and drying conditions, to complete the experimental protocol.

Comment 11. In addition to measuring water uptake, swelling ratio or dimensional changes (such as length or thickness) should be included to provide a clearer understanding of the membrane's structural stability.

Comment 12. The ED setup is incorrectly referenced in Figure 1a, which shows synthesis steps of the polymer. Please correct the figure reference.

Comment 13. In Figure 1a, DMC is mentioned without being defined. Please expand the abbreviation (e.g., dichloromethane or CH_2Cl_2) for clarity.

Comment 14. Please provide units for the variables C_t , C_i (initial and final molar concentrations), and time (t) in the relevant equations to ensure clarity and reproducibility.

Comment 15. Equation (4) states that “ C_t and C_i represent the initial and final molar concentrations,” but by convention, C_t typically denotes the concentration at time t (i.e., final concentration), and C_i refers to the initial concentration. Please revise this for clarity.

Comment 16. Radar plots in Figure S3 lack error bars and numerical precision. Please specify how many times each experiment was repeated. A comparative table with actual values and standard deviations would significantly improve the clarity of data.

Comment 17. The manuscript states that “after surface modifications, the IEC of the membranes decreased from NML-Q1-NH₂ to NML-Q3-NH₂,” but Figure S3 shows NML-Q3-NH₂ with a higher IEC. Please clarify this inconsistency or restate the sentence.

Comment 18. The manuscript mentions that “COOH groups serve as active functional sites, reducing IEC through electrostatic interactions with the oppositely charged QA groups.” However, while the IEC of NML-Q1-NH₂ is lower than that of NML-Q1, the IEC of NML-Q2-NH₂ and NML-Q3-NH₂ is higher than their respective non-modified counterparts. Please address this discrepancy and make a clear explanation for the relationship between surface modification and ion exchange capacity.

Comment 19. NML-Q3-NH₂ exhibits higher resistance (R_m) than NML-Q3 despite having higher IEC and water uptake. Is this due to surface modification? Cross-section SEM image should be provided to show the thickness of the ISIP-formed layer and clarify this point. Additionally, thicknesses of each membrane are recommended to be tabulated before and after surface modification.

Comment 20. The section “Optimizing ion transport channels” discussing ion transport mechanisms in traditional IEMs is redundant in this manuscript, as the focus is on anion permselectivity. Consider moving this background to Supplementary Information (e.g., under Supplementary Figure S4) to streamline the main text and remove repetitive statements.

Comment 21. Figure 4i does not depict hydration energies as described in the manuscript; it shows flux and selectivity. Please clarify.

Comment 22. High water uptake with high number of quaternary ammonium groups typically favor multivalent over monovalent ions, as hydrophilicity of the membranes increase. How do the authors explain the higher Cl⁻ selectivity over SO₄²⁻ in NML-Q2 and NML-Q3 membranes? Water contact angle measurements are strongly recommended to justify surface hydrophilicity/hydrophobicity as being stated in the manuscript and clarify the interaction of ions with varying hydration energies.

Comment 23. The permselectivity results are very impressive and surpass many reported in the literature. It is noteworthy that the high concentration gradient used (0.1 M NaCl + Na₂SO₄ vs 0.01 M KNO₃) likely enhances monovalent ion transport, making it difficult to distinguish the true contribution of membrane properties or it is together with process conditions. To ensure a fair and accurate comparison, the authors should report the testing conditions (current density and solution concentrations) for each study in Table S3. This clarification is necessary to properly assess the exceptional performance of the NML-Q2-NH₂ membrane.

Comment 24. The energy consumption calculations per unit of ions separated should be included along with the permselectivity values in Table S3, as this is a critical metric for potential scalability concerns.

Comment 25. It is worth adding the performance results obtained for Cl/SO₄ (>100) in the following study “Journal of Membrane Science 716 (2025) 123518” in Table S3.

Comment 26. The manuscript states “For example, SO₄²⁻, with a larger Stokes radius (2.31 Å), faces more resistance compared to Cl⁻ (1.21 Å).” Please provide reference for this information.

Comment 27. Please provide a separate table listing Stokes radii and hydration energies for all ions studied with corresponding references for readability and easier comparison of the permselectivity values.